# Unique properties of a subset of human pluripotent stem cells with high capacity for self-renewal

Kevin X. Lau[1,12], Elizabeth A. Mason [1,2,12], Joshua Kie[1], David P. De Souza [3], Joachim Kloehn[4], Dedreia Tull[3], Malcolm J. McConville [3,4], Andrew Keniry[5,6], Tamara Beck[5], Marnie E. Blewitt [5,6], Matthew E. Ritchie [5], Shalin H. Naik [5], Daniela Zalcenstein [5], Othmar Korn[7], Shian Su[5], Irene Gallego Romero[8], Catrina Spruce[9], Christopher L. Baker[9], Tracy C. McGarr [9], Christine A. Wells [1,2,10] & Martin F. Pera[1,5,9,11✉]

Archetypal human pluripotent stem cells (hPSC) are widely considered to be equivalent in developmental status to mouse epiblast stem cells, which correspond to pluripotent cells at a late post-implantation stage of embryogenesis. Heterogeneity within hPSC cultures complicates this interspecies comparison. Here we show that a subpopulation of archetypal hPSC enriched for high self-renewal capacity (ESR) has distinct properties relative to the bulk of the population, including a cell cycle with a very low G1 fraction and a metabolomic profile that reflects a combination of oxidative phosphorylation and glycolysis. ESR cells are pluripotent and capable of differentiation into primordial germ cell-like cells. Global DNA methylation levels in the ESR subpopulation are lower than those in mouse epiblast stem cells. Chromatin accessibility analysis revealed a unique set of open chromatin sites in ESR cells. RNA-seq at the subpopulation and single cell levels shows that, unlike mouse epiblast stem cells, the ESR subset of hPSC displays no lineage priming, and that it can be clearly distinguished from gastrulating and extraembryonic cell populations in the primate embryo. ESR hPSC correspond to an earlier stage of post-implantation development than mouse epiblast stem cells.

[1] Department of Anatomy and Neuroscience, University of Melbourne, Melbourne, Victoria 3010, Australia. [2] Centre for Stem Cell Systems, Department of Anatomy and Neuroscience, University of Melbourne, Melbourne, Victoria 3010, Australia. [3] Metabolomics Australia, Bio21 Institute of Molecular Science and Biotechnology, University of Melbourne, Parkville, Victoria 3052, Australia. [4] Department of Biochemistry and Molecular Biology, Bio21 Institute of Molecular Science and Biotechnology, University of Melbourne, Parkville, Victoria 3052, Australia. [5] Division of Molecular Medicine, The Walter and Eliza Hall Institute, 1G Royal Parade, Parkville, Victoria 3052, Australia. [6] Department of Medical Biology, University of Melbourne, Melbourne, Victoria 3010, Australia. [7] Australian Institute for Bioengineering and Nanotechnology, The University of Queensland, Brisbane, Queensland 4072, Australia. [8] Melbourne Integrative Genomics, School of Biosciences, University of Melbourne, Melbourne, Victoria 3010, Australia. [9] The Jackson Laboratory, Bar Harbor, ME 04609, USA. [10] Divisions of Cancer and Hematology and Molecular Medicine, The Walter and Eliza Hall Institute, 1G Royal Parade, Parkville, Victoria 3052, Australia. [11] The Florey Institute of Neuroscience and Mental Health, 30 Royal Parade, Parkville, Victoria 3052, Australia. [12]These authors contributed equally: Kevin X. Lau, Elizabeth A. Mason. ✉email: martin.pera@jax.org

The successful application of human pluripotent stem cells (hPSC) in research and cell therapy relies on the ability to maintain, expand, and differentiate these cells in vitro in a tightly controlled and efficient fashion. Our understanding of the regulation of pluripotent stem cell self-renewal and lineage specification is in turn built largely on embryological paradigms, with developmental roadmaps providing critical knowledge of the key transitional stages, and pinpointing the extrinsic and internal molecular pathways drive cell fate decisions. In the mouse, we now have a fairly clear understanding of the states of pluripotency that span the developmental stages between the blastocyst and the late gastrula in vivo[1]. The characterization of mouse naive embryonic stem cells (ESC)[2] and epiblast stem cells (EpiSC)[3,4] as in vitro equivalents of the preimplantation epiblast and the anterior primitive streak, respectively[5], has shed considerable light on the properties of the cultured cells. Recently, a stage between these two pluripotent states called formative pluripotency, corresponding to the early post-implantation epiblast, has been described[6]. Defined by specific molecular and biological features, including an absence of lineage priming and a rapid response to induction of lineage specification (including the germline lineage), formative mouse pluripotent stem cells have yet to be successfully serially propagated in vitro.

The derivation of mouse EpiSC, their characterization as epithelial cells dependent upon activin and FGF2 for maintenance in the pluripotent state[3,4], and their co-expression of lineage-specific and pluripotency genes[4,5,7–10], led many researchers to the conclusion that hPSC derived and maintained under conventional culture conditions (archetypal hPSC), which share these features, equate to the primed state of mouse pluripotency. This in turn led to a search for conditions that would support long-term maintenance of hPSC in a naive state[11,12]. Several culture systems have been described that support a cell with molecular features quite similar to the human preimplantation epiblast[13,14]. Extended propagation of diploid hPSC in these systems remains challenging, however.

Population heterogeneity complicates the interpretation of stem cell phenotype. To dissect heterogeneous populations in archetypal hPSC cultures, we used monoclonal antibodies to cell surface antigens to define subsets of stem cells that exist in a hierarchical continuum of cell states. When we subjected these cells to transcriptional profiling, we observed co-expression of pluripotency genes with lineage specific transcription factors, particularly in subpopulations of cells with lower levels of stem cell surface marker and pluripotency gene expression[15]. Importantly though, we also found that cells which expressed pluripotency markers at high levels were less likely to display lineage priming.

Subsequently by analyzing cells grown under defined conditions, and using a more refined sorting strategy to isolate the subpopulation enriched for high self-renewal (ESR) followed by medium throughput single-cell RT-QPCR, we were able to show that the cells at the top of the hierarchy expressed very uniform and high levels of pluripotency markers, and showed no lineage priming[16].

As cells traverse through stages of pluripotency in vitro and in vivo, they undergo changes in cell cycle regulation and metabolic activity, they restructure their epigenome, and their gene expression profile changes. The ability to sort highly purified populations of hPSC with high self-renewal capacity, and to analyse transcription in these subpopulations at the genomic level using RNA-Seq for subpopulation and single cell analysis, coupled with the availability of new single-cell gene expression data from preimplantation and post-implantation primate embryos, prompted us to re-examine the properties of the ESR subpopulation of archetypal hPSC, and to reconsider its developmental status. The results show that the ESR subpopulation resembles the primate early post-implantation epiblast, similar to the mouse formative state of pluripotency.

## Results

**Self-renewal of subsets of hPSC.** We have previously used flow cytometry to isolate cell subpopulations, followed by assay of colony forming ability as an indicator of self-renewal[15,17]. hPSC survival after dissociation to single cells and flow cytometry is poor. We therefore developed a simple methodology that would allow comparison of self-renewal of defined cell populations at reasonable levels of initial survival after flow cytometry, through sorting small aggregates of cells, to maintain cell–cell contacts and enhance post-sort survival[18].

Fluorescence activated cell sorting using antibodies GCTM-2 and TG30 (recognizing CD9) enabled us to recover small cell aggregates (chiefly doublets; singlets, 17%; doublets, 61%; triplets, 16%; quadruplets, 6%) (Fig. 1a). We seeded wells with equal cell numbers as aggregates or singlets. Although both aggregates and single cells attached to the plate after sorting, a much larger fraction of the aggregates had begun to spread 1 h after plating, indicative of high viability; the difference was more evident after 24 h (Fig. 1b). Initial attachment, spreading and survival of the GCTM-2$^{high}$CD9$^{high}$ subpopulation was similar to the GCTM-2$^{mid}$CD9$^{mid}$ subpopulation. Flow cytometry re-analysis of both subpopulations 72 h later showed that they had largely retained their cell surface phenotypes though as shown previously[19], the GCTM-2$^{high}$CD9$^{high}$ population had begun to reconstruct the entire cell state continuum (Fig. 1c). By 4 days, cells plated as aggregates displayed a higher colony forming efficiency than single cells, and the GCTM-2$^{high}$CD9$^{high}$ subpopulation had formed a much larger number of microcolonies (Fig. 1d–e). Colonies formed from GCTM-2$^{high}$CD9$^{high}$ aggregates were larger and showed a higher proportion of cells bearing stem cell markers compared with colonies formed by GCTM-2$^{mid}$CD9$^{mid}$ cells (Fig. 1f). Time-lapse video microscopy confirmed that the initial cell numbers attaching to the dish were similar for the two subpopulations. However, subsequent monitoring showed that while both subpopulations were migratory and underwent cell division, the GCTM-2$^{high}$CD9$^{high}$ subpopulation persisted to form colonies of 4–32 cells several days later, whereas the GCTM-2$^{mid}$CD9$^{mid}$ colonies suffered abortive expansion and, in many cases, extinction (Supplementary Movies 1 and 2).

**Differentiation potential of hPSC subpopulations.** In the mouse, the ability to differentiate efficiently into germ cells in vitro is limited to epiblast like cells at the stage of formative pluripotency[20,21]. Naive pluripotent stem cells or epiblast stem cells both lack this capacity[21]. Using the two-step protocol to generate first a mesoderm-like intermediate and subsequently convert these cells to PGC-like cells[22], we measured the degree of differentiation of the GCTM-2$^{high}$CD9$^{high}$EPCAM$^{high}$ and GCTM-2$^{mid}$CD9$^{mid}$ populations using flow cytometry to quantitate EPCAM/ITGA6 double high cells which represent PGC-like cells (Fig. 2a). GCTM-2$^{high}$CD9$^{high}$EPCAM$^{high}$ cells showed higher expression of EPCAM and ITGA6 at the onset, as expected. After 2 days of differentiation into mesoderm-like cells, both groups showed loss of EPCAM expression but at Day 4 after PGC-like cell induction, an EPCAM/ITGA6 double positive fraction was observed in both groups. Cultures that did not receive growth factors to induce germ cell formation contained no EPCAM/ITG6A double high population. The GCTM-2$^{high}$CD9$^{high}$EPCAM$^{high}$ population had largely disappeared by the end of the time course, as noted by Sasaki et al.[22]. The identity of the PGC-like cells was confirmed by staining with antibodies to PRDM1

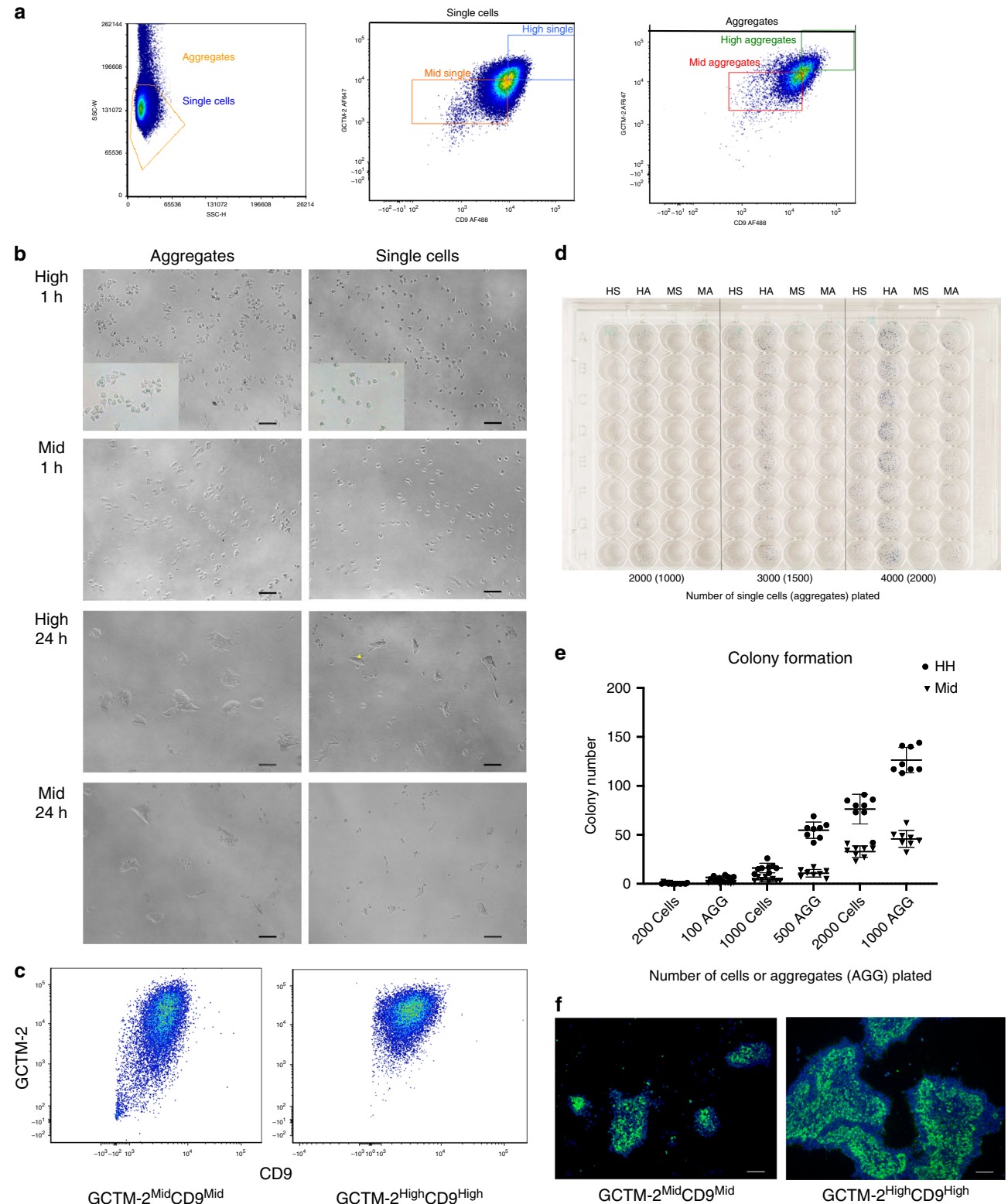

and NANOS3 (Fig. 2b). More PGC-like cells were consistently obtained from GCTM-2[high]CD9[high]EPCAM[high] compared with GCTM-2[mid]CD9[mid] subpopulations in both cell lines, though there was a high degree of inter-assay variability (Fig. 2c). We and others have shown previously that the ESR fraction of hPSC is pluripotent (Figs. 4 and 5, refs. [17,19]). In this study, directed differentiation in adherent culture, carried out on cell lines WA09 and WA01, confirmed that both the GCTM-2[high]CD9[high]

subpopulation and the remaining cell population were pluripotent, as measured by expression of markers characteristic of progenitors of all three embryonic germ layers (Fig. 2d).

**ESR cells have a cell cycle with a low G1 fraction**. Initial analyses of the mouse ESC cell cycle indicated that naive mouse ESC, and cells of the epiblast, show a shortened cell cycle with a minimal

**Fig. 1 Assay of self-renewal of subpopulations of hPSC under conditions that maintain cell–cell contacts. a** Isolation of aggregates of subsets of hPSC by flow cytometry. Left panel, side and forward scatter; middle panel, separation of single cells using GCTM-2 and anti-CD9; right panel, separation of cell aggregates using GCTM-2 and anti-CD9. **b** Phase contrast image of aggregates and single cells 1 and 24 h after plating. At 1 h, both populations of aggregates (GCTM-2$^{high}$CD9$^{high}$, HIGH and GCTM-2$^{mid}$CD9$^{mid}$, MID) have attached and spread onto the substrate. Scale bar = 100 micron. **c** Flow cytometry analysis of cell surface antigen expression in aggregate cultures prepared as in a 72 h after plating. **d** Microcolony formation after 4 days by single cells (200, 1000, or 2000) or aggregates (100, 500, or 1000) of GCTM-2$^{high}$CD9$^{high}$ (HS, HA) and GCTM-2$^{mid}$CD9$^{mid}$ (MS, MA) subpopulations. **e** Numbers of microcolonies formed at 4 days by GCTM-2$^{high}$CD9$^{high}$ or GCTM-2$^{mid}$CD9$^{mid}$ single cells or aggregates (AGG). Values represent the mean ± standard deviation from eight wells from one experiment; see Supplementary Table 5 for biological replicates of this assay. **f** Immunostaining with stem cell surface marker antibody GCTM-2 of colonies formed by aggregates of GCTM-2$^{mid}$CD9$^{mid}$ and GCTM-2$^{high}$CD9$^{high}$ subpopulations. Scale bar = 100 micron. Results in **b**, **d**, and **f** display representative outcomes from three experiments.

G1 component[23]. While more recent work shows that the cell cycle state of naive mouse stem cells is dependent upon culture conditions[24], it remains clear that lineage commitment is coupled to cell cycle regulation in mouse and human PSC, and that the G1 phase represents a decision point for undergoing lineage specification[25–27].

Here we re-assessed our previous finding that a subset of cells expressing high levels of the stem cell antigen GCTM-2 had a reduced G1 fraction compared with the remainder of the population[28]. Using a more refined sorting procedure and EdU incorporation to identify S-phase cells. we determined the cell cycle phase distribution of GCTM-2$^{high}$CD9$^{high}$EPCAM$^{high}$, GCTM-2$^{high}$CD9$^{high}$, and GCTM-2$^{low}$CD9$^{low}$ subpopulations, compared with unsorted cells (the general population). As shown in Fig. 3, very few of the GCTM-2$^{high}$CD9$^{high}$EPCAM$^{high}$ or GCTM-2$^{high}$CD9$^{high}$ cells were in G0/G1 phase, with most of this subpopulation in S or G2/M. By contrast, GCTM-2$^{low}$CD9$^{low}$ cells were predominantly in G0/G1 (~70%). The cell cycle phase distribution of the unsorted general population was consistent with these findings. Results using a different WA09 subline with a FUCCI reporter[29] confirmed these conclusions (Supplementary Fig. 1).

**Active mitochondria and bivalent metabolism in ESR cells.** Metabolic activity is modulated throughout early mammalian development. The preimplantation relies on a combination of aerobic glycolysis and oxidative phosphorylation, and this metabolic pattern is maintained during post-implantation development up to E7.5 in the mouse[30]. Like the epiblast, naive state mouse ESC show this bivalent metabolism, while primed epiblast stem cells rely primarily on glycolysis[31]. Similar transitions from bivalent metabolism to glycolysis have been reported during conversion of naive to primed hPSC[32,33].

To determine if the energy metabolism status of hPSC is dependent on their position in the pluripotency hierarchy, we first assessed the mitochondrial membrane potential of defined subsets of hPSC by measuring the uptake of the mitochondrial dye tetramethylrhodamine methyl ester (TMRM), or the combined uptake and the ratio of red/green fluorescence of the dye JC-1. Live cell staining of WA09 cells showed that staining with TMRM was strongest at the edge of the colonies, where cells expressing the highest levels of the stem cell marker CD9 are found (Fig. 4a). To quantitate mitochondrial activity across the cell populations, WA09 cells were incubated with TMRM or JC-1, labeled with stem cell surface makers GCTM-2, TG30 (anti-CD9), and anti-EPCAM, and the GCTM-2$^{high}$CD9$^{high}$EPCAM$^{high}$ and GCTM-2$^{low}$CD9$^{low}$ fractions were identified by flow cytometry. The GCTM-2$^{high}$CD9$^{high}$EPCAM$^{high}$ subpopulation stained more intensely with TMRM, and showed higher ratio of red to green fluorescence following incubation with JC-1 compared with the general (remaining) population, or the GCTM-2$^{low}$CD9$^{low}$ subpopulation (Fig. 4b–c), indicating increased mitochondrial activity in the GCTM-2$^{high}$CD9$^{high}$EPCAM$^{high}$ subpopulation.

Analysis of mitochondrial oxidative phosphorylation in live cells using the Agilent Seahorse XF apparatus confirmed that the basal oxygen consumption rate (OCR) was higher in the GCTM-2$^{high}$CD9$^{high}$EPCAM$^{high}$ subpopulation compared with the remaining GEN population (Fig. 5a). These cells also exhibited higher spare mitochondrial capacity, as indicated by maximum OCR achieved after addition of proton uncoupling agent FCCP (Fig. 5a). Comprehensive analysis of intracellular metabolite levels in the GCTM-2$^{high}$CD9$^{high}$EPCAM$^{high}$ cells and unfractionated population using liquid chromatography and gas chromatography–mass spectrometry (LC–MS, GC–MS) provided further evidence that the GCTM-2$^{high}$CD9$^{high}$EPCAM$^{high}$ cells exist in a distinct metabolic state. Principal Component Analysis (PCA, Supplementary Fig. 2a, b) and hierarchical cluster analysis (Fig. 5b, LC–MS; Supplementary Fig. 2c, GC–MS) clearly separated the purified cell population from the GEN population. Consistent with the live cell staining and OCR analysis, the GCTM-2$^{high}$CD9$^{high}$EPCAM$^{high}$ contained elevated levels of TCA cycle metabolites (Fig. 5c) and was depleted in many amino acids and metabolites in the urea cycle (Supplementary Tables 1–2), compared with the general population. Pathway analysis further confirmed the distinct metabolic profile of the GCTM-2$^{high}$CD9$^{high}$EPCAM$^{high}$ subpopulation (Supplementary Fig. 2d, e).

To gain further insights into the metabolic wiring of this subpopulation of cells, GCTM-2$^{high}$CD9$^{high}$EPCAM$^{high}$ fractionated cells and the GEN population of hPSC were cultivated in the presence of $^{13}$C glucose for 2 h and level of $^{13}$C-enrichment in different intermediates of central carbon metabolism monitored by GC–MS. High levels of $^{13}$C-enrichment were observed in all intermediates in glycolysis and the pentose phosphate pathway confirming that both cell populations exhibit high rates of aerobic glycolysis (Fig. 5d). Interestingly, $^{13}$C-enrichment in serine and glycine, which are synthesized from the glycolytic intermediate 3-phosphoglycerate, were higher in the GCTM-2$^{high}$CD9$^{high}$EPCAM$^{high}$ cells, indicating that these cells may have higher rates of amino acid and protein synthesis. Consistent with GCTM-2$^{high}$CD9$^{high}$EPCAM$^{high}$ cells having elevated rates of mitochondrial metabolism, $^{13}$C-enrichment in citrate and isocitrate were substantially upregulated in this subpopulation (Fig. 5d). However, $^{13}$C-enrichment in later intermediates in the oxidative cycle (succinate, fumarate, and malate) were not significantly changed between the two cell populations indicating that early intermediates in the TCA cycle may be diverted into anabolic pathways (catapleurosis). Increased catapleurosis in the GCTM-2$^{high}$CD9$^{high}$EPCAM$^{high}$ cells was supported by the elevated levels of $^{13}$C-labeling in glutamate, which is synthesized from the TCA cycle intermediate, α-ketoglutarate, as well as high levels of $^{13}$C-enrichment in long chain unsaturated fatty acids (i.e., oleic acid) and cholesterol, indicating high rates of membrane biogenesis or turnover. Overall, these data provide compelling evidence that the GCTM-2$^{high}$CD9$^{high}$EPCAM$^{high}$ cells are metabolically more active than the GEN hPSC population,

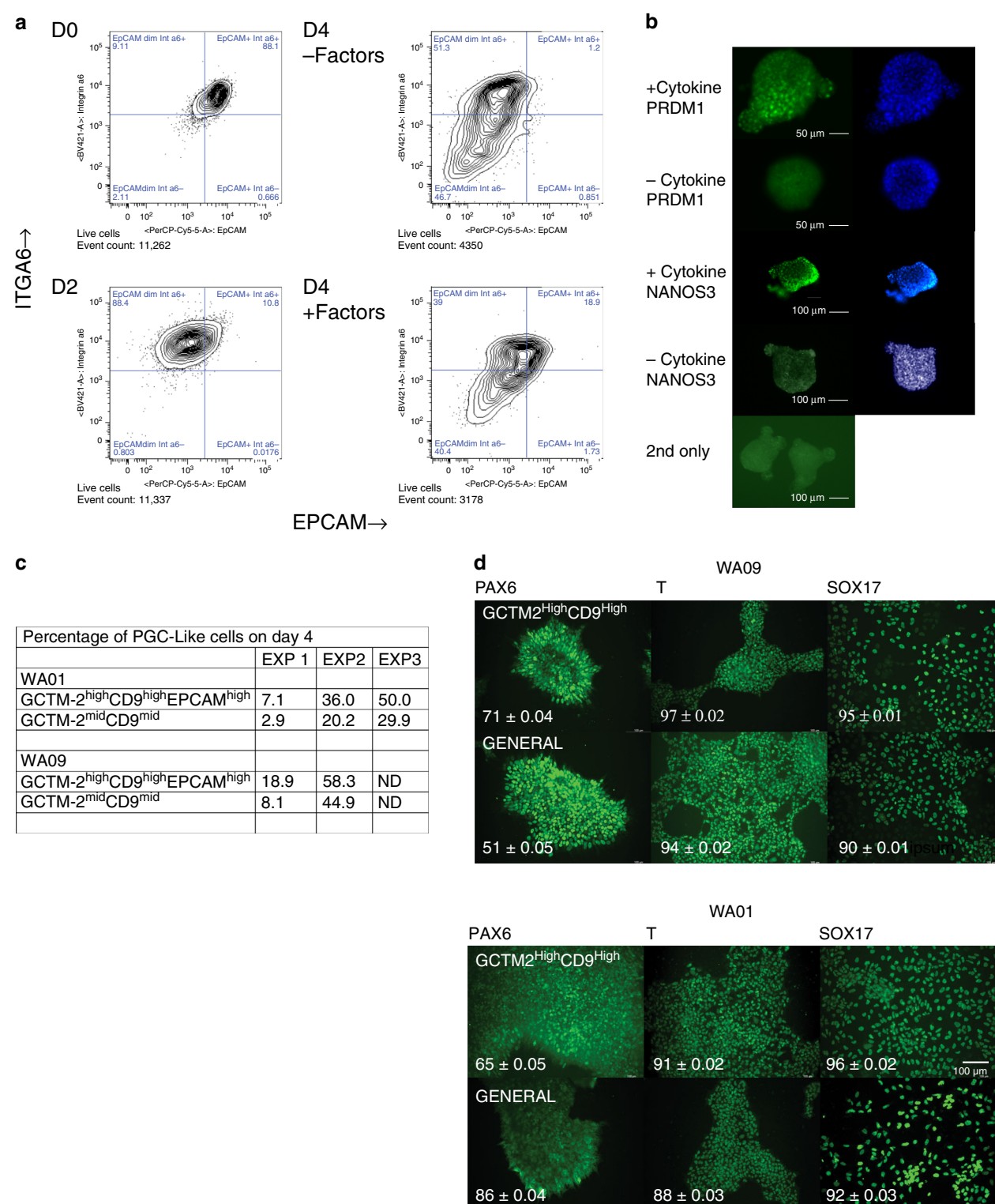

| Percentage of PGC-Like cells on day 4 | | | |
|---|---|---|---|
| | EXP 1 | EXP2 | EXP3 |
| **WA01** | | | |
| GCTM-2high CD9high EPCAMhigh | 7.1 | 36.0 | 50.0 |
| GCTM-2mid CD9mid | 2.9 | 20.2 | 29.9 |
| | | | |
| **WA09** | | | |
| GCTM-2high CD9high EPCAMhigh | 18.9 | 58.3 | ND |
| GCTM-2mid CD9mid | 8.1 | 44.9 | ND |
| | | | |

exhibiting higher rates of oxidative phosphorylation and anabolic amino acid and lipid synthesis.

**DNA methylation in hPSC cultured under defined conditions.** Levels of DNA methylation in the mouse and human epiblast are low, but increase substantially in the mouse following embryo implantation and activation of the de novo methyltransferases *Dnmt3a* and *Dnmt3b*[34,35]. Reduced representation bisulfite sequencing to assess levels of DNA methylation showed no major differences between the relatively low mean levels of overall DNA

methylation or levels of DNA methylation over CpG islands between the GCTM-2high CD9high EPCAMhigh subpopulation and the general population (Fig. 6a, b). DNA methylation distributions were bimodal (Fig. 6c, d), and there was little evidence of differential methylation at CpG islands across any particular loci in the GCTM-2high CD9high EPCAMhigh subpopulation (Fig. 6e). The extent of DNA methylation at CpG islands in various repeat elements was high and did not vary between the unsorted and GCTM-2high CD9high EPCAMhigh fraction (Fig. 6f). De novo DNA methyltransferases *DNMT3A* and *DNMT3B* were both expressed

**Fig. 2 Differentiation potential of GCTM-2$^{high}$CD9$^{high}$EPCAM$^{high}$ and GCTM-2$^{mid}$CD9$^{mid}$ subpopulations. a** Flow cytometry assay showing differentiation of GCTM-2$^{high}$CD9$^{high}$EPCAM$^{high}$ subpopulation into PGC-like cells. Panels show flow cytometry analysis for expression of EPCAM and ITGA6 in starting population (Day 0), post induction of incipient mesoderm-like cell with ACV and CHIR99021 (Day2), and PGC induction (addition of BMP4, LIF, KITLG, and EGF) versus controls without these factors (Days 4 + and − factors). **b** Staining of aggregates of PGC-like cells for PRDM1 or NANOS3 on Day 4. Aggregates incubated with cytokines showed strong nuclear staining; those incubated in the absence of factors did not. DNA staining to right of each image. Scale bar PRDM1 panels 50 μM; NANOS3 and 2nd antibody only, 100 μM. Staining with secondary antibody alone in bottom panel. **c** Table showing percentage yield of EPCAM$^+$ITG6A$^+$ cells on Day 4 in two cell lines for GCTM-2$^{high}$CD9$^{high}$EPCAM$^{high}$ and GCTM-2$^{mid}$CD9$^{mid}$ subpopulations. **d** Directed differentiation of the GCTM-2$^{high}$CD9$^{high}$ fraction and the remaining population of WA09 and WA01 cells in adherent culture. Panels show staining for PAX6, T, and SOX17 after induction of differentiation for 5 days. Numbers on each panel represent the proportion of cells positive for the indicated marker ± 95% confidence interval. Scale bar = 100 μM, same magnification in all panels. Results in **b** and **d** display representative outcomes from two experiments on two cell lines.

along with *TET1* across the cell populations studied (below and Supplementary Fig. 3a–j), similar to the early post-implantation epiblast in the mouse[36], and suggestive of a highly dynamic state of DNA methylation in these cells.

**Differential chromatin accessibility in hPSC subpopulations.** We next identified critical differences in the chromatin landscape between GCTM-2$^{high}$CD9$^{high}$EPCAM$^{high}$ and GCTM-2$^{mid}$CD9$^{mid}$ populations using the assay for transposase accessible chromatin[37]. In total, we identified 118,442 regions as accessible across both populations. Of these, 3144 were more accessible in the GCTM-2$^{high}$CD9$^{high}$EPCAM$^{high}$ cells, while 4730 were more accessible in the GCTM-2$^{mid}$CD9$^{mid}$ population (Fig. 7a, FDR < 0.01; quality assessment data Supplementary Fig. 4, Supplementary Data 1). Generally, peaks with increased accessibility in the high population were distant from transcription start sites (TSS), more often found in intergenic and introns typical of enhancers, whereas peaks with increased accessibility in the middle populations were closer to TSS (Fig. 7b) identified as promoters (Supplementary Fig. 5a). To annotate open chromatin sites in each population we compared their location to the entire set of transcription factor (TF) binding sites across a diverse range of human cell types identified by the ENCODE project[38] using a locus overlap enrichment analysis[39] (Supplementary Data 2). Genomic locations more accessible in the GCTM-2$^{high}$CD9$^{high}$EPCAM$^{high}$ population were highly enriched for TF binding sites identified in hPSC compared with the GCTM-2$^{mid}$CD9$^{mid}$ population (Fig. 7c). Furthermore, the most enriched TFs binding at the GCTM-2$^{high}$CD9$^{high}$EPCAM$^{high}$ population sites included known pluripotency factors NANOG, POU5F1 (OCT4), and TCF12 and BCL11A (Fig. 7d, upper panel), the latter two TF having been previously identified as highly expressed in the GCTM-2$^{high}$CD9$^{high}$EPCAM$^{high}$ population[16]. While these same factors were still identified as enriched in the regions in the GCTM-2$^{mid}$CD9$^{mid}$ populations, other general chromatin factors showed the highest overlap (Fig. 7d, lower panel). Finally, we compared regions of increased accessibility in both populations to the tissue and cell-type specific DNAse hypersensitivity clusters identified in human samples[40]. Again, this analysis identified that the regions with greatest enrichment in the GCTM-2$^{high}$CD9$^{high}$EPCAM$^{high}$ population were annotated as being unique to stem cells (Supplementary Fig. 5b, c). In summary, these data indicate that the GCTM-2$^{high}$CD9$^{high}$EPCAM$^{high}$ population has greater open chromatin at putative enhancers bound by canonical pluripotent factors at sites unique to stem cells, whereas the low population has greater DNA accessibility at promoters bound by general chromatin and transcription factors. Together these data support the hypothesis that the chromatin of cells in the GCTM-2$^{high}$CD9$^{high}$EPCAM$^{high}$ population exists in a distinct state compared with the low population.

**Comparison of ESR cell transcriptome with primate epiblast.** RNA-seq analysis comparing gene expression in the GCTM-2$^{high}$CD9$^{high}$EPCAM$^{high}$ subpopulation to the general (total unfractionated) population (Supplementary Data 3) identified 515 genes differentially expressed between the GCTM-2$^{high}$CD9$^{high}$EPCAM$^{high}$ subset and the total population (132 upregulated in the GCTM-2$^{high}$CD9$^{high}$EPCAM$^{high}$ cells and 383 downregulated relative to the unfractionated cells, at >1.5-fold change in expression level with an adjusted *p* value <0.05; Fig. 8a).

Genes upregulated in the GCTM-2$^{high}$CD9$^{high}$EPCAM$^{high}$ fraction included *NODAL* and its antagonists *LEFTY1* and *LEFTY2*, in agreement with our previous study[16]. *POU3F1*, a marker of the naive to formative transition in the mouse, was also upregulated. Notably, a number of small nuclear and small nucleolar RNAs were expressed at high levels in the GCTM-2$^{high}$CD9$^{high}$EPCAM$^{high}$ population. Negative regulators of MAPK signaling, including *DUSP5* (inactivator of ERK1), *DUSP6* (inactivator of ERK2), and *SPRY2*, were upregulated in the self-renewing fraction, as was *DACT1* (an antagonist of canonical WNT signaling). Negative regulators of MAPK signaling including *Dusp4* and *Spry* were recently shown to be upregulated at an early stage during dissolution of the mouse naive state[41]. Amongst the genes expressed at lower levels in the GCTM-2$^{high}$CD9$^{high}$EPCAM$^{high}$ cellular subset relative to the unfractionated population were members of the WNT signaling pathway, including *WNT4*, *FRZB*, *FZD3*, *FZD5*, and *FZD8*. Of the top 100 genes upregulated in the general population, 47 genes (all upregulated at twofold change or higher) were previously reported to be expressed at peak levels at the onset of neural differentiation in the CORTECON study of Temple and colleagues[42] (global analysis, Supplementary Fig. 6). Examination of previously published microarray data[43] for a subset of these neural induction genes confirmed a pattern of continuous upregulation in cell subsets with decreasing levels of pluripotency associated cell surface markers in multiple cell lines (Supplementary Fig. 7, data visualized in the Stemformatics platform https://www.stemformatics.org). *BMP2*, *BMP4*, and *FST* were also upregulated in the general population, consistent with our previous results[16].

Some genes characteristic of the primate preimplantation epiblast and naive hPSC were expressed in the GCTM-2$^{high}$CD9$^{high}$EPCAM$^{high}$ fraction (PRDM14, TFCP2L1, ZFP42, DPPA2, and TFAP2C), but others were not (ARGFX, KLF17, TBX3, NLRP7). Genes expressed in primitive endoderm (SOX17, GATA4, GATA6, FOXA2, and APOA2) were not found in either the GCTM-2$^{high}$CD9$^{high}$EPCAM$^{high}$ or the general population, nor were genes activated during early gastrulation (T, MIXL, GSC, EOMES, FOXA2, LHX1).

scRNA-seq on hPSC fractionated into four separate subpopulations enabled us to compare gene expression in the fractionated subpopulations with the single-cell data of Nakamura et al.[44] for *Macaca fascicularis* pre- and post-implantation embryos (quality

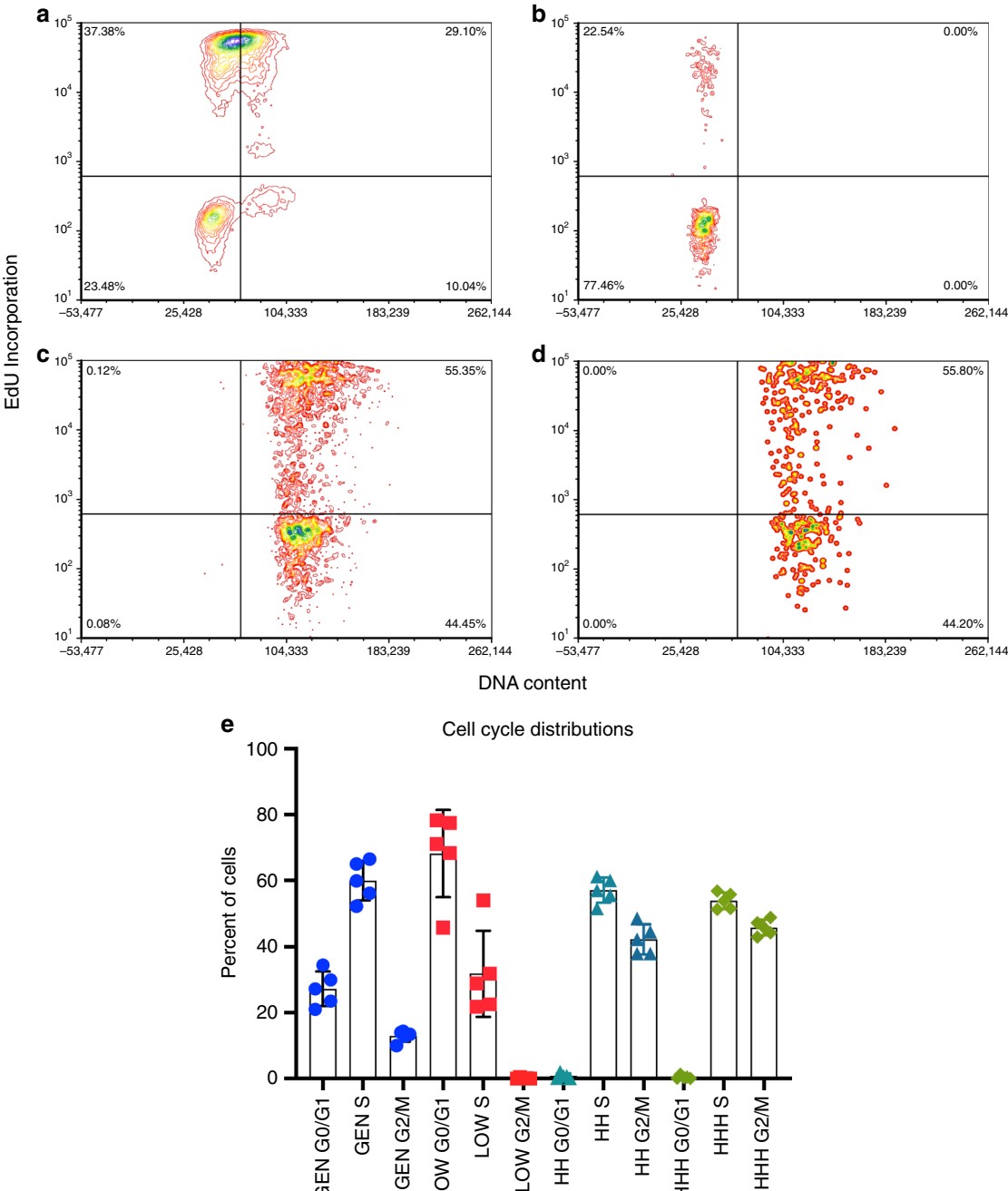

**Fig. 3 Cell cycle analysis of hPSC subpopulations by flow cytometry of EdU labeled cultures.** Cells were pulsed labeled with EdU, sorted into three subpopulations using cell surface markers GCTM-2 and CD9 with or without EPCAM, and analyzed by flow cytometry. **a–d** Flow cytometry profiles from one experiment. **a** Unsorted population; **b** GCTM-2$^{low}$CD9$^{low}$; **c** GCTM-2$^{high}$CD9$^{high}$; **d** GCTM-2$^{high}$CD9$^{high}$EPCAM$^{high}$; **e** summary showing results from seven experiments. Mean values are shown and error bars represent standard error of seven biological replicates; General is unsorted population, low is GCTM-2$^{low}$CD9$^{low}$, HH is GCTM-2$^{high}$CD9$^{high}$, and HHH is GCTM-2$^{high}$CD9$^{high}$EPCAM$^{high}$. See Supplementary Table 6 for biological replicate data points.

assessment, Supplementary Figs. 8–9). We analyzed 300 cells, and we detected expression of 8403 genes. All cells uniformly expressed the general pluripotency associated transcription factors POU5F1, SOX2, and NANOG (Supplementary Fig. 10a–d)). ZFP42, a marker of the naive state in mouse, was expressed throughout the population, but another naive state marker, TFCPL1 was expressed primarily in the GCTM-2$^{high}$CD9$^{high}$EP-CAM$^{high}$ and GCTM-2$^{high}$CD9$^{high}$ subsets; POU3F1, character-istic of post-implantation epiblast, was expressed throughout (Supplementary Fig. 10e–g). PCA of the human cell

subpopulations alone indicated that they could be clearly separated along a continuum of cell states (Fig. 8b). Ontology analysis of differential gene expression highlighted a number of pathways involving ribonucleoprotein complexes, ribosomes, and metabolic processes, in concurrence with the expression of small nuclear and nucleolar RNAs noted above in the subpopulation analysis (Supplementary Table 3), and a number of pathways related to oxidative metabolism, including metabolic processes, mitochondrion, mitochondrion organization, generation of

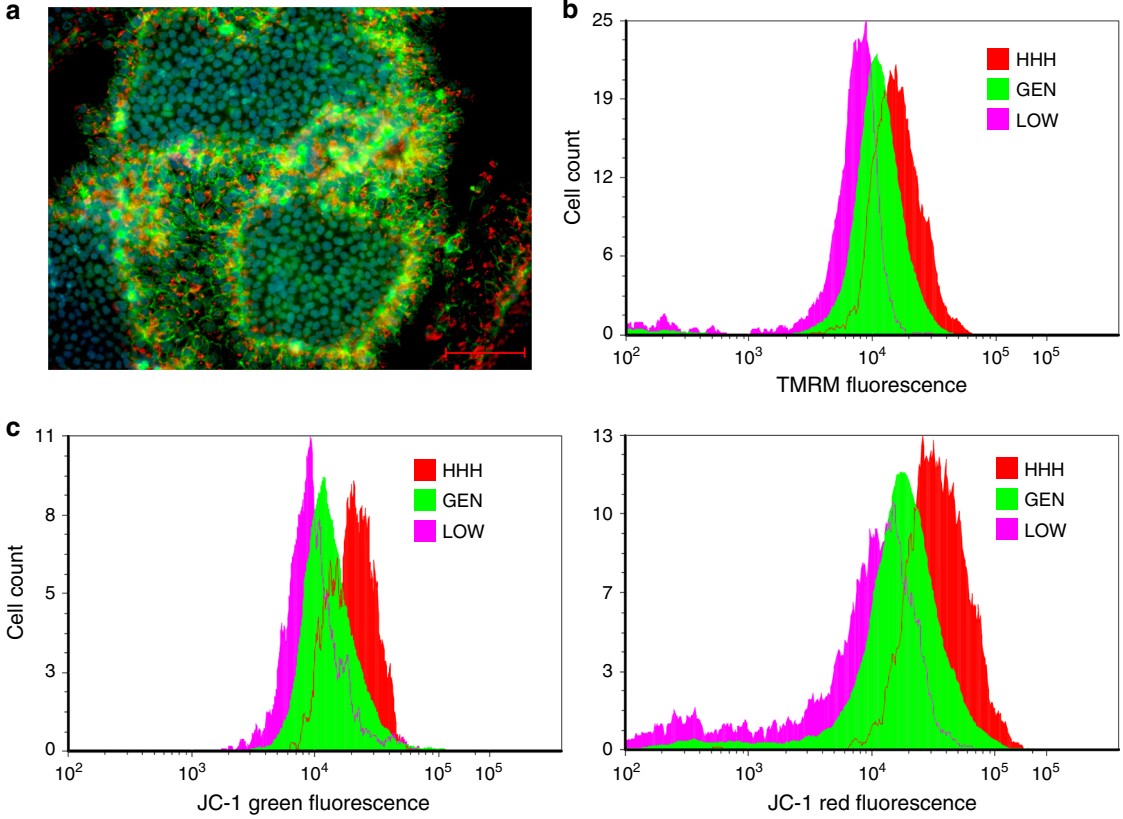

**Fig. 4 Mitochondrial activity in subpopulations of hPSC. a** Double label staining of live cells with TMRM and stem cell surface marker CD9. Cells at the edges of colonies stain most strongly with antibody and dye. Scale bar = 100 micron. **b** Flow cytometry analysis of TMRM staining in GCTM-2^highCD9^highEPCAM^high (HHH), GCTM-2^lowCD9^low (LOW), and unsorted (GEN) population. **c** Flow cytometry analysis of green (left panel) and red (right panel) JC-1 dye emission in GCTM-2^highCD9^highEPCAM^high (HHH), GCTM-2^lowCD9^low (LOW), and unsorted (GEN) population. GCTM-2^highCD9^highEPCAM^high cells display highest ratio of green to red emission, indicative of high mitochondrial membrane potential. Results in a display representative outcomes from three experiments.

precursor metabolites and energy, mitochondrial inner membrane, and hydrogen ion transmembrane transporter activity.

The GCTM-2^highCD9^highEPCAM^high subset and general populations both expressed markers of the post-implantation mouse epiblast, including *POU3F1*, *OTX2*, *DNMT3A*, *DNMT3B*, *SOX4*, *SOX11*, *LIN28A*, and *ZNF281* (Supplementary Data 3). In a PCA of the human single-cell data and the cynomolgus data of Nakamura et al.[44] (Fig. 8c), the first principal component resolved the two experiments with cynomolgus and human cells. The second principal component resolved the inner cell mass and pre- and post-implantation epiblast, along with extraembryonic cells. In this dimension, the human cells aligned with preimplantation epiblast stage cells. In the third principal component, which separated post-implantation stages of cynomolgus development, human cells aligned with the post-implantation epiblast, with cells in the GCTM-2^highCD9^highEPCAM^high fraction between early and late post-implantation stages. The human cells were clearly distinguished from inner cell mass and preimplantation epiblast stages, from gastrulating cells, and from extraembryonic tissues. PCA of the cynomolgus cells alone revealed that it was difficult to separate early from late post-implantation epiblast (Supplementary Fig. 11).

The cells used in this study were WA09 cells grown on mTeSR in feeder-free, serum-free conditions. To assess the generality of these findings, we compare these results with two previous microarray analyses: an independent study that examined similarly defined subpopulations of cell lines MEL1 and WA09 grown in proprietary serum replacement with mouse embryo fibroblast feeders or mTeSR1 (WA09)[43], and our previous study using cell line ES02 grown in serum-supplemented medium in the presence of mouse fibroblast feeder cells[17]. We identified a panel of stage-specific genes (Supplementary Table 4) on the basis of their differential expression in the data of Nakamura et al.[44] and a recent scRNA-seq study of the human preimplantation embryo[45] (Fig. 9). Genes specific to the inner cell mass, or mainly expressed in the inner cell mass and preimplantation epiblast, were very weakly expressed in the GCTM-2^highCD9^high subpopulation in the previous works and in the GCTM-2^highCD9^highEPCAM^high cells in the current RNA-seq study. Expression levels of the gene panel characteristic of the inner cell mass, preimplantation epiblast and early post-implantation epiblast were found at appreciable levels in self-renewing hPSC in all studies. Those genes with highest expression levels in all three cynomolgus epiblast populations, or in early and late post-implantation epiblast, were expressed robustly in all hPSC subpopulations in all studies. The gene panel specific to late post-implantation epiblast and gastrulating populations was expressed at somewhat lower levels in all of our populations relative to pan-epiblast specific genes. Last, genes characteristic of gastrulation stages 2A and 2B (nomenclature, ref. [44]) were expressed at low levels in the GCTM-2^high-CD9^highEPCAM^high and GCTM-2^highCD9^high and GCTM-2^mid-CD9^mid subpopulations, with levels rising in the GCTM-2^lowCD9^low and GCTM-2^−CD9^− fractions. Thus, these data support the same conclusion as PCA of the single cell data: hPSC cell subpopulations enriched for self-renewal capacity show a pattern of gene expression that is strongly similar to early

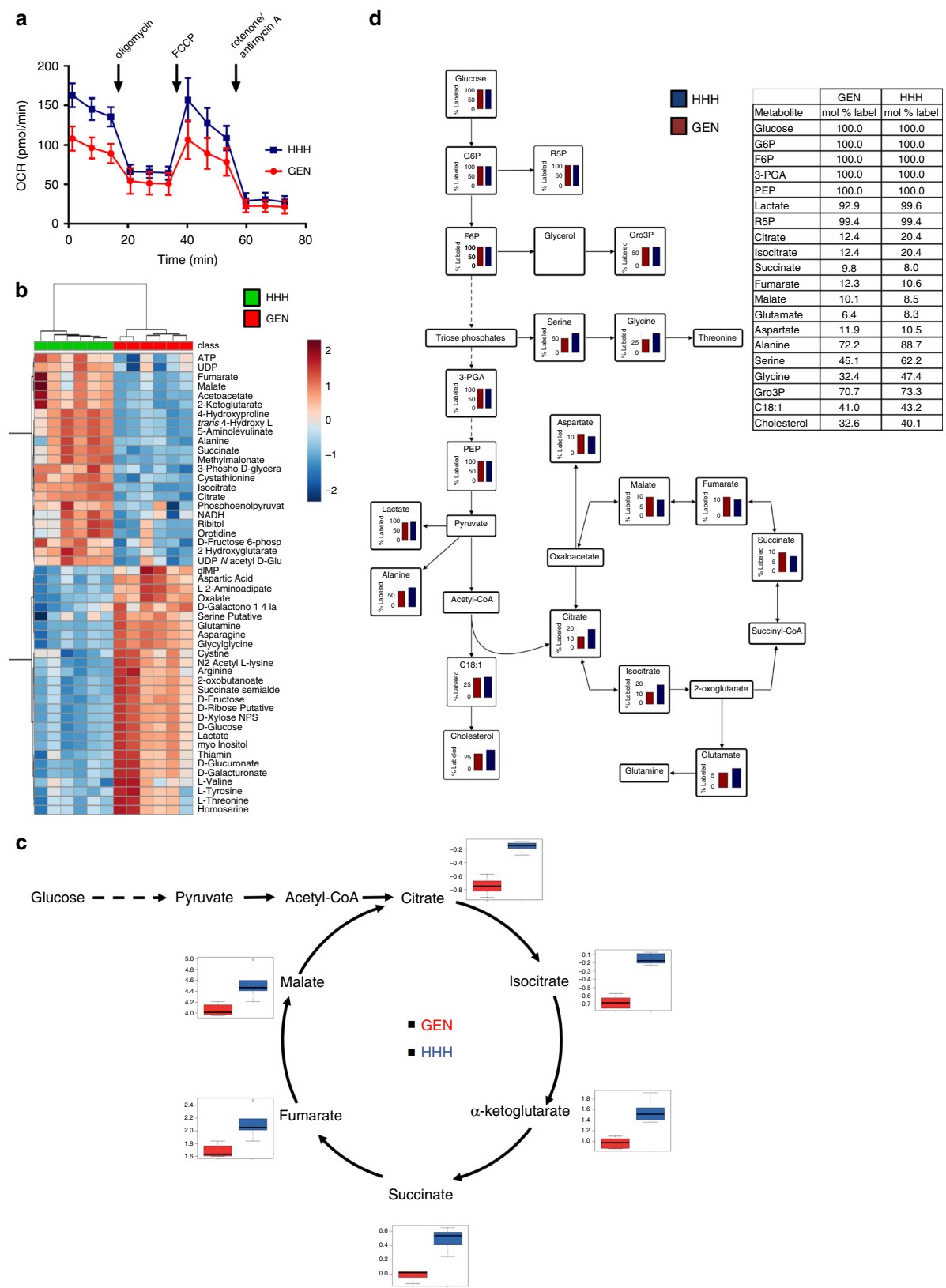

post-implantation epiblast stages in the primate embryo, but clearly distinguished from inner cell mass or gastrulation stages.

## Discussion

We showed previously that the minority ESR subpopulation of hPSC can be isolated with cell surface markers and identified by colony formation assay. Colony formation measures both survival and self-renewal, and dissociation to single cells and flow cytometry compromises survival. We have used several assay strategies to avoid conflation of survival and self-renewal[16,17], but the approach described in this study of isolating aggregates is simple, and yields defined subpopulations with high initial survival for

**Fig. 5 Analysis of the metabolism of GCTM-2^highCD9^highEPCAM^high cells. a** Seahorse XF analysis of oxygen consumption rate (OCR) in the GCTM-2^highCD9^highEPCAM^high (HHH) and unsorted (GEN) cell populations. Data represent means ± standard error. Differences in the measurements of basal respiration and respiration after oligomycin, FCCP, and rotenone/antimycin between the HHH and general cell population were compared in a two-talied *t*-test; the *p*-values was 0.0000031, 0.0025, 0.0032, and 0.005, respectively, for six biological replicates per condition with three measurements each. **b** Metabolomic analysis of GEN and HHH cells. Unsupervised hierarchical cluster analysis of metabolite levels determined by LC–MS differing between the two samples (*t*-test with Benjamini–Hochberg FDR = 0.05) in replicate samples leads to clear separation of the two populations. The metabolite abundance values are normalized by scaling to zero mean and unit variance by compound. Cells colored red denote higher abundance, while blue denotes lower abundance. **c** Levels of TCA cycle intermediates are elevated in GCTM-2^highCD9^high cells. Polar metabolites from unfractionated hPSC and GCTM-2^highCD9^high cells were analyzed by GC–MS and relative abundance of TCA cycle intermediates after medium normalization shown as box plots. Values are means ± standard error of three biological replicates. For each boxplot, the bisecting line of each box represents the median. The top and bottom ends of each box are the 75th and 25th percentiles, respectively. The top and bottom horizontal lines extending out of each box are the minimum and maximum values, respectively. **d** HHH and GEN cells were metabolically labeled with ^13C glucose for 2 h and ^13C-enrichment (expressed as mol%) in select intermediates measured by GC–MS.

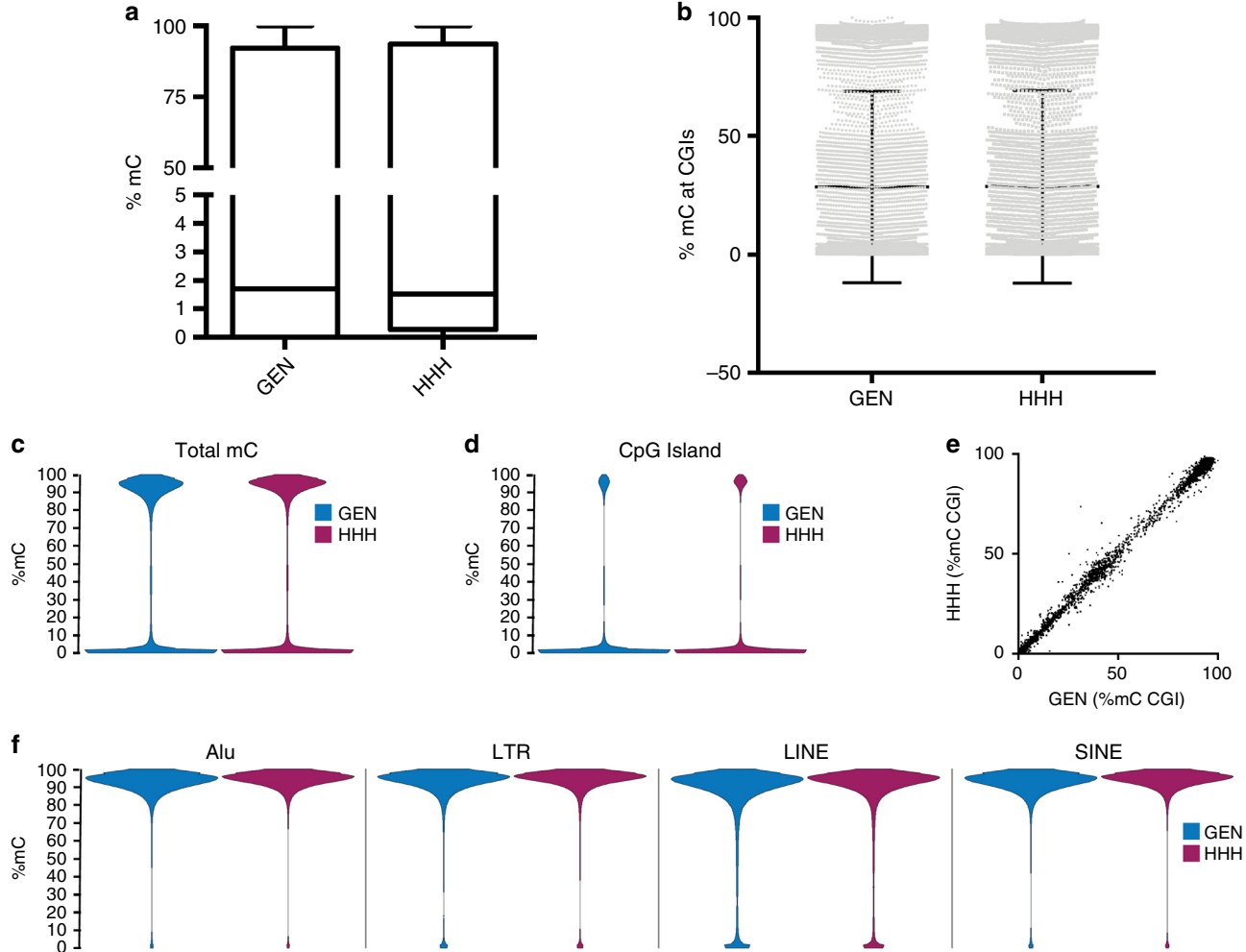

**Fig. 6 Reduced representation bisulfite sequencing analysis of DNA methylation in unsorted (GEN) and GCTM-2^highCD9^highEPCAM^high (HHH) subpopulations. a, b** Box plots of overall DNA methylation (**a**) and methylation at CpG islands (**b**). In **a**, line indicates median, box shows 25th to 75th percentile, and bars show maxima and minima. In **b**, line shows the mean and error bars show standard deviation. **c, d** Bean plots showing the distribution of DNA methylation (%mC) of individual CpGs, both genome wide (**c**) and at CpG islands (**d**). **e** Scatter plot showing the %mC at all CpG islands comparing the GEN and HHH populations. **f** Bean plots showing the %mC of individual CpGs at the repetitive elements of type Alu, LTR, LINE, and SINE. All data shown is the average for GEN (*n* = 2) and HHH (*n* = 3).

subsequent analysis. Although the initial survival of aggregates of the hPSC subpopulations was similar, further development of viable stem cell colonies was observed predominantly in the fraction bearing the highest level of stem cell markers. Time-lapse video microscopy revealed that cells in the GCTM-2^highCD9^high fraction formed microcolonies that persisted during extended propagation, whereas microcolonies of cells in the lower fraction underwent gradual extinction. This is similar to the observations of Barbaric et al.[46], who found that only a subset of SSEA3-positive hPSC formed microcolonies that persisted and grew. Their findings and ours suggest that self-renewal might be a function of the formation of a critical mass of cells expressing

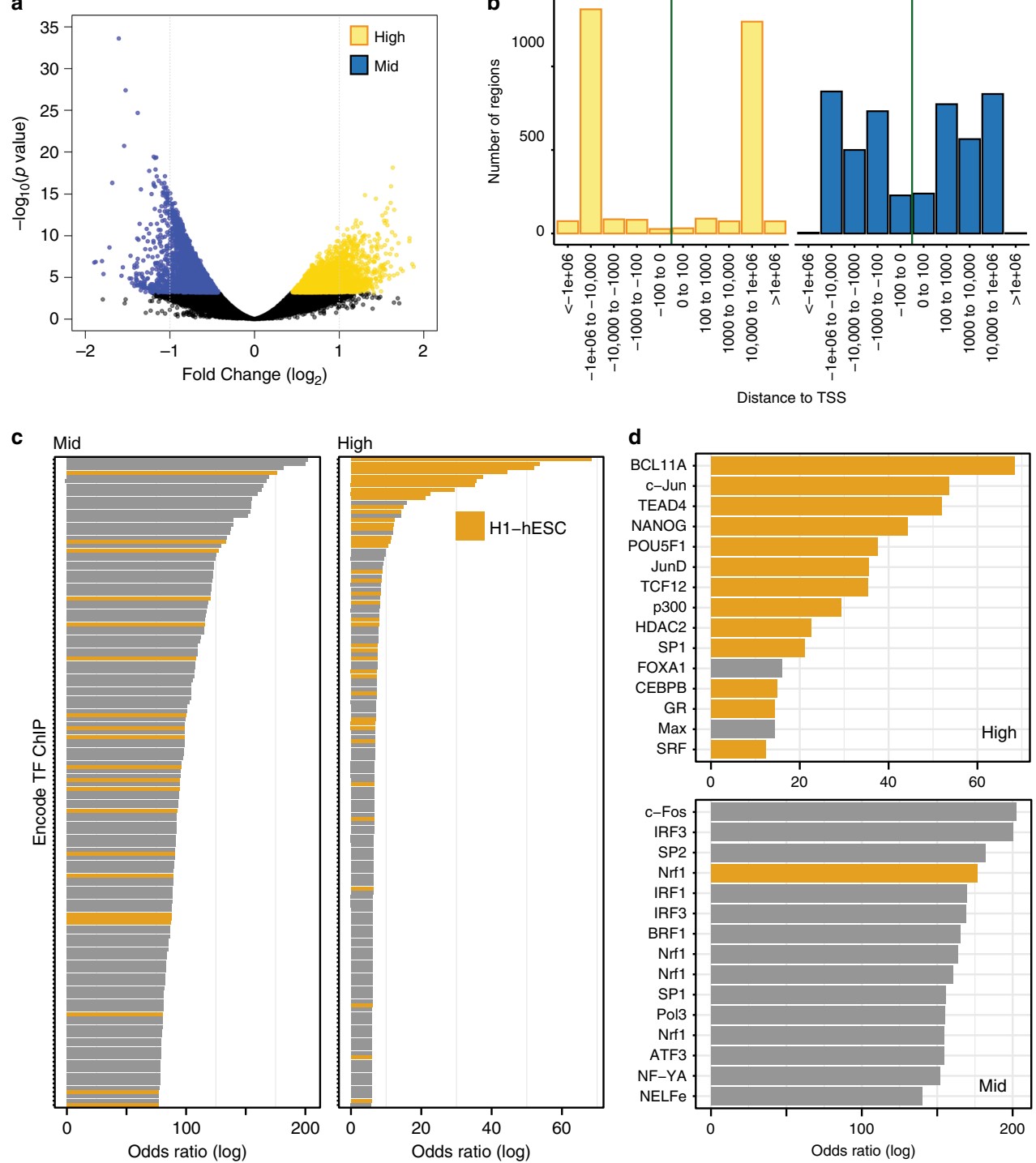

**Fig. 7 Landscape of accessible chromatin differentiates stem cell populations. a** Volcano plot showing significant differences in DNA accessibility as measured using ATAC-seq between GCTM-2highCD9highEPCAMhigh (yellow) and GCTM-2midCD9mid (blue) populations (FDR < 0.01). Vertical lines indicate twofold difference. **b** Distribution of significantly different open chromatin regions plotted as distance from TSS. **c** Bar chart of the top 150-log odds ratios for overlap of differentially open chromatin regions in the GCTM-2midCD9mid (left) and GCTM-2highCD9highEPCAMhigh (right) populations compared with all ChIP-seq data for all cell types in the ENCODE TF dataset. Orange indicates ChIP data from the human ESC cell line H1. **d** Similar to **c** showing to 15 highest log odds ratios for overlap. Names of each TF antibody used for ChIP are indicated on the left.

survival or growth factors at a sufficiently high local level. We have shown that cells in the GCTM-2highCD9high or GCTM-2highCD9highEPCAMhigh fractions express the highest levels of components of the NODAL signaling pathway[16,17]. Recent results in zebrafish indicate that cell–cell contacts are key to Nodal signaling[47], suggesting the possibility that hPSC might similarly

depend on a positive feedback loop of NODAL signaling and cell–cell adhesion to drive self-renewal.

In the mouse, under appropriate culture conditions, naive cells pass through an intermediate state between naive and primed pluripotency, and in this transient state, are competent to undergo germline differentiation[21]. It was previously reported

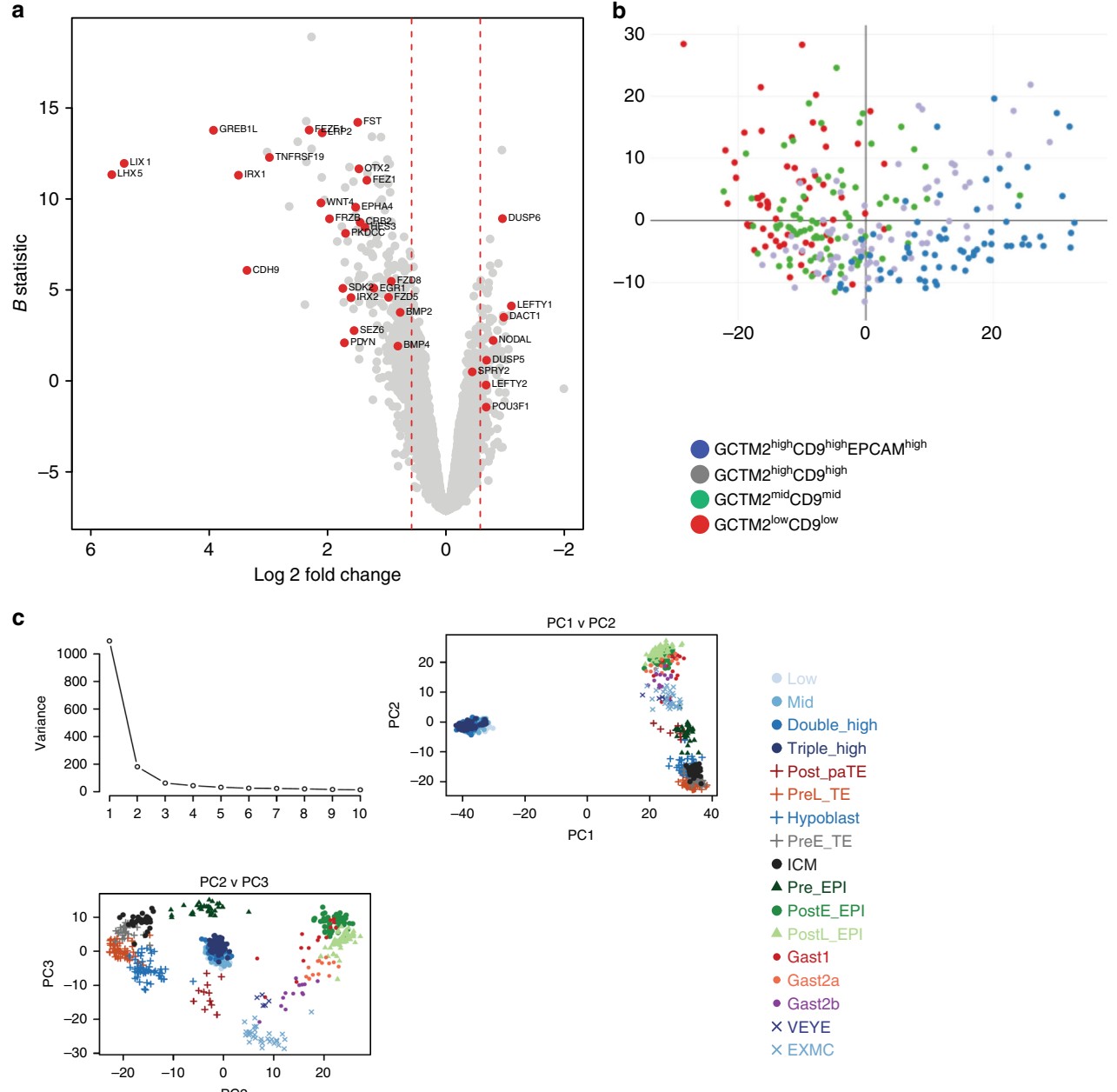

**Fig. 8 Global gene expression analysis of hPSC subpopulations by RNA-seq. a** Volcano plot illustrating differentially expressed genes in GCTM-2highCD9highEPCAMhigh versus general (unsorted) populations. **b** Principal component analysis of single-cell RNA-seq data on GCTM-2highCD9highEPCAMhigh, GCTM-2highCD9high, GCTM-2midCD9mid, and GCTM-2lowCD9low subpopulations. **c** Joint species principal component analysis of single-cell RNA-seq data on GCTM-2highCD9highEPCAMhigh (HHH), GCTM-2highCD9high (HH), GCTM-2midCD9mid (MID), and GCTM-2lowCD9low (LOW) subpopulations alongside cynomolgus embryo data from ref. [30]; single embryo cells classified according to Houghton et al.[30]. Top left: screeplot demonstrating the amount of variability in the data accounted for by each component; top right: graph displaying data distribution along first and second components; bottom left: graph displaying data distribution along the second and third components. Color and shape of point indicate sample phenotype, each point representing a single cell.

that archetypal hPSC can form primordial germ cell-like cells[22]. We confirm this finding and show that ESR hPSC have the capacity for germline differentiation. This distinguishes these cells from the naive and primed states in the mouse.

Cells in the GCTM-2highCD9highEPCAMhigh fraction displayed a cell cycle with a very limited G1 fraction relative to other cells in the population. It has been shown that hPSC pause in G1 when preparing to embark on differentiation[25,26,28]. It is possible that ESR stem cells do not execute such a differentiation checkpoint and continue in a self-renewing loop, until a pause in the cell cycle is activated, possibly through an RB-dependent mechanism[48].

Cells in the GCTM-2highCD9highEPCAMhigh fraction show higher mitochondrial membrane potential and increased oxidative phosphorylation compared with the general population. Comprehensive metabolite profiling and [13]C-glucose labeling studies confirmed increased rates of catabolism of pyruvate in the mitochondrial TCA cycle. These studies also showed that increased TCA cycle flux in the GCTM-2highCD9highEPCAMhigh cells is used to generate key intermediates including citrate and α-ketoglutarate which are subsequently exported from the mitochondria and used for synthesis of lipids and amino acids/proteins. Elevated rates of amino acid synthesis in the GCTM-2highCD9highEPCAMhigh cells

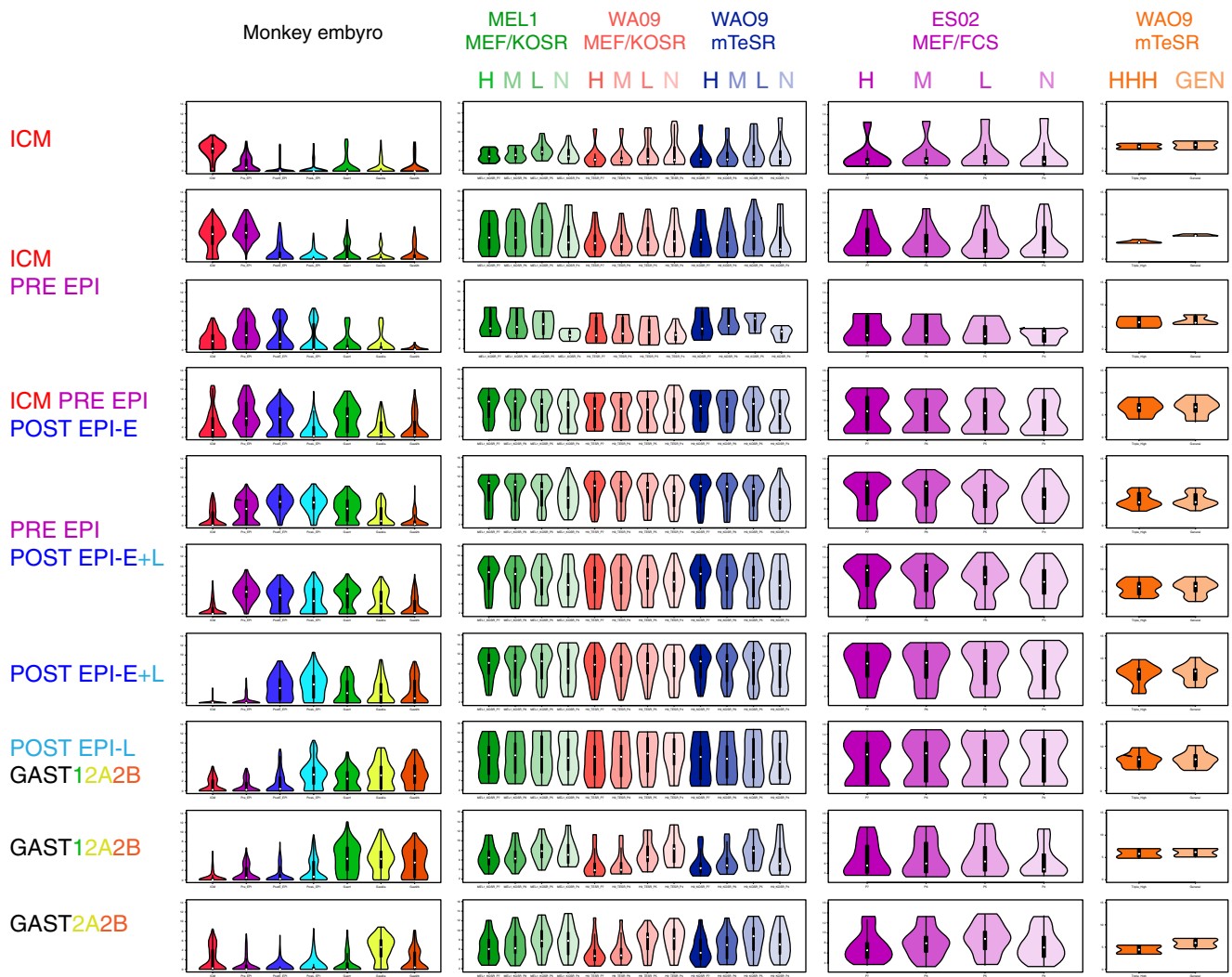

**Fig. 9 String section plot comparison of expression of embryonic stage-specific genes by scRNA-seq in cynomolgus embryo[30] with expression in subpopulations of hPSC.** The violin plots in the left hand column show expression from single-cell RNA-seq data in ref. [30], with classification of cells as described therein; embryonic stages in which particular gene sets (Supplementary Table 4, each row in the figure displays violin plots for one gene set) are expressed are listed to the left of the plots. The second column shows microarray data from ref. [27] for the same sets of genes in GCTM-2[high]CD9[high] (H), GCTM-2[mid]CD9[mid] (M), GCTM-2[low]CD9[low] (L), and GCTM-2[neg]CD9[neg] (N) subpopulations of MEL1 cells grown in the presence of medium containing Knockout Serum Replacer on mouse embryo fibroblast feeder cell layers (MEF/KSOR), or WA09 cells grown in MEF/KSOR or mTeSR defined medium. The third column shows microarray data from ref. [16] for the same sets of genes on GCTM-2[high]CD9[high] (H), GCTM-2[mid]CD9[mid] (M), GCTM-2[low]CD9[low] (L), and GCTM-2[neg]CD9[neg] (N) subpopulations of ES02 cells grown in serum-containing medium on mouse embryo fibroblast feeder cell layers (MEF/FCS). The last column shows RNA-seq bulk data on the same sets of genes from the current study for GCTM-2[high]CD9[high]EPCAM[high] (HHH) and unsorted (GEN) cells. Colors of violins correspond to embryonic stages or cell subpopulations identified in labels at left or top of figure (last three columns), respectively.

was also supported by increased flux of glycolytic intermediates into serine and glycine synthesis. The high rate of de novo synthesis of cholesterol in hPSC is unusual and further highlights the importance of TCA cycle in generating precursors (i.e., citrate) for this pathway. Overall, these data indicate that the GCTM-2[high]CD9[high]EPCAM[high] cells retain a highly active anabolic metabolism.

DNA methylation levels in mouse naive ES are generally low, similar to the pattern in the preimplantation epiblast, and rise in vivo as development progresses toward gastrulation. In the mouse or primate post-implantation embryo, *DNMT3A* and *DNMT3B*, and *TET1* are activated shortly after implantation[36]. Levels of methylation in the GCTM-2[high]CD9[high]EPCAM[high] and general population were similar in this study, and considerably lower than levels reported for mouse epiblast stem cells. In human hPSC, co-expression of DNA methyltransferases and TET

enzymes could account for the dynamic nature of DNA methylation, exemplified by the dramatic response of these cells to the presence of ascorbic acid, an activator of TET enzymes, in the cell culture medium[49].

In contrast to DNA methylation levels, chromatin accessibility varied strikingly in the cell subpopulations that we studied. Chromatin regions that showed high accessibility in the GCTM-2[high]CD9[high]EPCAM[high] mapped to sites of previously identified pluripotency transcription factor binding in hPSC, and to sites of DNAse hypersensitivity that were found to be unique to stem cells. This pattern changed markedly in the GCTM-2[mid]CD9[mid] population. These findings suggest that targets of pluripotency transcription factors in the GCTM-2[high]CD9[high]EPCAM[high] cell fraction might be involved in the regulation of self-renewal. We found no evidence for enrichment of binding sites for TFAP2C in open chromatin in our GCTM-2[high]CD9[high]EPCAM[high] cells,

thus distinguishing this population from the naive state of hPSC described earlier[50].

The properties of the ESR subpopulation of hPSC discussed above are consistent with those of the mouse early post-implantation epiblast. The ability of archetypal hPSC to undergo differentiation into amnion[51–53] or the germline[22,54] is also consistent with a phenotype closer to an earlier post-implantation state rather than to the EpiSC. We have confirmed the capacity of archetypal hPSC for germline differentiation, a capacity which is found in the majority of the population of archetypal hPSC grown in defined conditions; further study with refinements to this assay will be required to confirm a higher capacity for germline differentiation in GCTM-2[high]-CD9[high]EPCAM[high] cells. We previously used embryoid body assays[16] to show that the ESR subpopulation is pluripotent, as did another study using the teratoma assay[19]. These findings are not surprising, since flow cytometry analyses of replated populations of the ESR subpopulation reported here (Fig. 1a vs. 1c) and in a more extensive previous work[19] indicate that this subpopulation can eventually regenerate the entire hierarchy of archetypal hPSC. Here quantitative assessment of directed differentiation into three germ layer lineages in adherent cultures confirmed that both the ESR and the remaining population are pluripotent. Together, our current data and the previous work are consistent with a developmental equivalence of the ESR with early post-implantation epiblast, and not naive or primed pluripotent states.

Previous studies have indicated that gene expression in archetypal hPSC aligns with the primate post-implantation but not preimplantation epiblast[44,45]. Our studies on the gene expression in the subpopulation of hPSC with a high capacity for self-renewal are consistent with these findings. The post-implantation epiblast persists longer in the primate than in the mouse, and it is difficult to resolve changes in epiblast gene expression during the post-implantation period until gastrulation, when pluripotency genes turn off and the activation of lineage-specific programs occurs. Our results here and elsewhere demonstrate that the cell cycle and metabolic profile, and the lack of lineage priming in the archetypal hPSC subpopulation showing a high capacity for self-renewal, are all consistent with early post-implantation epiblast, but not an mouse EpiSC-like state. Whether a subpopulation within mouse EpiSC cultures with high self-renewal capacity exists is unknown.

Recently Nakanishi et al.[55] also described a subpopulation of hPSC with high self-renewal capacity. These cells, which reside at the periphery of hPSC colonies, were isolated on the basis of cell surface expression of NCAD, and display priming toward the primitive endoderm lineage. We previously identified a subset of cells at the periphery of hPSC colonies which co-isolated with the self-renewing subpopulation and similarly co-expressed pluripotency and primitive endoderm genes (GATA4/GATA6/HNF4A/BMP2/FN1)[16]. We found that these cells were more abundant in cultures grown in medium supplemented with serum replacement on a feeder cell layer relative to defined conditions; they were quite rare in the HSR subpopulation isolated from cultures grown in defined medium[16]. We did not observe cells expressing markers of primitive endoderm in the current study. It is possible that these lineage primed cells are descendants of the subpopulation we describe here. Determination of the relationship between the NCAD[+] cells and the HSR fraction we have identified will require further analysis, but it is clear that the majority of HSR cells in hPSC cultures grown under defined conditions resemble the early post-implantation epiblast, and not primitive endoderm. Cornacchia et al.[56] recently reported that hPSC cultured in E8 media display features that are intermediate between naive and primed hPSC. We have used E8 and mTeSR interchangeably in these studies and have not noted differences between the two with respect to the parameters we have studied. Several features of the E8 cultures described by Cornacchia et al.[56] including increased capacity for self-renewal, pluripotency associated marker and gene expression, and the metabolome, are similar to the properties of the ESR population we describe. A simple interpretation of both sets of data and consistent with prior observations[16,43] is that culture in defined media shifts the overall population toward the ESR state.

The self-renewing subpopulations of archetypal hPSC that we have identified here and elsewhere represent a minority of the culture with distinct properties. The majority of the archetypal hPSC population shows features closer to mouse EpiSC, as noted by others. Even under the defined culture conditions used in this study that enhance self-renewal, transcriptional evidence of neural lineage priming is evident in the general population. The development of methods to propagate pure populations of self-renewing archetypal hPSC populations in a state similar to the formative stage in mouse might enhance the efficiency of cloning and differentiation protocols, and reduce the variability in differentiation efficiency often observed between hPSC lines from different genetic backgrounds. Further elucidation of the molecular regulatory pathways that maintain cells in this state will enhance our understanding of human development and guide efforts to model embryology in a dish.

## Methods

**hPSC culture, differentiation, and marker expression**. Experimental procedures for culture and differentiation of human embryonic stem cells, indirect immuno-fluorescence microscopy and flow cytometry followed minor modifications to established protocols.

**hPSC culture**. Human embryonic stem cell stocks (WA09 and FUCCI-G1) were maintained as described previously[57]. The FUCCI cells were a gift from Prof Jonathan S. Draper[29]. Routine maintenance of hPSC was carried out in serum-supplemented medium with fibroblast feeder cell support and subculture was performed using Dispase or collagenase to harvest fragments of colonies dissected manually.

For experiments, cells were transferred to defined feeder-free conditions (mTeSR conditions) using either mTeSR1 medium or Essential 8™ medium on Matrigel or recombinant vitronectin and subculture performed with Dispase or Tryple Express according to the manufacturers' protocols.

For dissociation to single cells, cultures were treated with 10 μM of InSolution™Blebbistatin (Merck Millipore cat. no. 203389) for 1 h prior to dissociation to single cells using TrypLE™ (Life Tech, cat. no. 12605). Cultures were incubated in media supplement with 10 μM of InSolution™Blebbistatin overnight after which Blebbistatin was removed.

Routine tests confirmed the absence of mycoplasma contamination and a diploid karyotype (20/20 cells on G-banding) in the cell lines used.

**Fluorescence activated cell sorting for colony formation assays**. Cells grown for 24 h in the presence of Y-27632 Rho kinase inhibitor were dissociated using Accutase, harvested, and examined under the microscope to ensure that most of the cells had not completely dissociated into single cells. Immunolabeling was carried out by incubation in a primary antibody cocktail containing TG30 anti-CD9 antibody (mouse IgG2a) and GCTM-2 (mouse IgM) (neat supernatants, this laboratory) for 20 min at 4 °C, followed by incubation in a secondary antibody cocktail containing goat anti-mouse IgG2a Alexa Fluor 488 antibody and goat anti-mouse IgM Alexa Fluor 647 antibody (A21131, A21238, both from Thermofisher), diluted in 2% FBS in DMEM-F12 flow cytometry buffer, again at 4 °C for 20 min. The cells were resuspended the cells in 1x DAPI in cold mTeSR1 before filtering the suspension through a 35-μm cell strainer to remove any larger aggregates and debris. Commercial antibodies were used at the dilutions recommended by the manufacturer.

All cell sorting was performed on a BD FACSAria III, using the 100-μm nozzle at 17 psi. A custom cytometer configuration was created that used the 17 psi pressure as opposed to the typical 20 psi pressure. This was performed to minimize aggregates from dissociating into single cells as they were deposited into the 96-well plates. To isolate the GCTM-2[high]CD9[high] and GCTM-2[mid]CD9[mid] fractions of aggregates and single cells, the gating strategy in Fig. 1a was used (forward and side scatter were employed to gate out debris and to identify single cells and aggregates; propidium iodide or AAD-7 was used to identify dead cells). The various fractions were deposited into 96-well tissue culture plates coated with Matrigel via the automated cell deposition unit on the Aria III. Subsequent to sorting, tissue culture plates were centrifuged at 200 × g for 2 min at room temperature. The sheath solution and media containing mixture was discarded and replaced with 150 μL of mTeSR1 medium.

Single cells from each of two FACS populations (GCTM-2$^{high}$CD9$^{high}$ and GCTM-2$^{mid}$CD9$^{mid}$) were plated at a density between 100 and 4000 cells per well of a 96-well Matrigel-coated plate. Aggregates from each of two FACS populations were placed between 50 and 2000 aggregates per well of a 96-well Matrigel-coated plate (Corning, Costar 3603). Cultures were maintained in mTESR1 with medium changed once after 48 h post-plating. After 72 h, wells were fixed and stained for colony counting as follows. The culture media was removed from cells and each well rinsed once with PBS prior to fixing with 100% ethanol for 10 min at room temperature. Ethanol was removed and wells allowed to air dry for 30 min prior to staining with haematoxylin for 10 min. Wells were then washed four times with Milli-Q water. In all, 0.08% aqueous ammonia solution was added into each well and incubated at room temperature for 2 min. After that, the wells were also washed three times with Milli-Q water and allowed to dry overnight. For time-lapse video microscopy, cells were imaged during 1–24 h post-plating in an environmental chamber (Clear State Solutions TCH 885-9G) to maintain a humid atmosphere with 5% CO$_2$ in air while phase contrast exposures were recorded every 15 min with a Leica SP8 Confocal microscope.

**Differentiation potential of hPSC subpopulations**. Differentiation into primordial germ cell-like cells was carried out essentially as described[22]. Cultures grown in defined, feeder-free conditions (mTeSR1 medium, above) were sorted into GCTM-2$^{high}$CD9$^{high}$ and GCTM-2$^{mid}$CD9$^{mid}$ fractions and plated onto fibronectin-coated dishes in Glasgow Minimal Essential Medium supplemented with Activin A (50 ng/ml), CHIR99021 (3 μM), and Y-27632 Rho kinase inhibitor (10 μM) for induction of incipient mesoderm-like cells. Two days later, cultures were harvested and replated as aggregates into Glasgow Minimal Essential Medium supplemented with LIF (10 ng/ml), BMP4 (200 ng/ml), KITLG (100 ng/ml), and EGF (50 ng/ml), or without these supplements in low attachment 96-well plates. Samples were taken at Day 0, 2, 4, and 6 and analyzed by flow cytometry for expression of cell surface ITGA6 and EPCAM using anti-human and mouse ITGA6 BV421 (Rat IgG2a, Biolegend 313624) and mouse anti-human CD236-PerCP (Mouse IgG2b, Biolegend 324213). On Day 4, some aggregates were fixed with 4% paraformaldehyde in PBSA for 30 min, permeabilized in 0.2% TritonX100 and 10% bovine serum albumin in PBSA for 30 min, stained overnight with rabbit antisera against PRDM1 (rabbit monoclonal IgG clone 9115 from Cell Signaling Technology) or NANOS3 (rabbit antisera, Abcamab70001), then incubated 1 h with goat anti-rabbit IgG Alexa488 conjugate, and counterstained with DAPI (Life Tech, Cat. #D1306), prior to examination under indirect immunofluorescence microscopy. Commercial antibodies were used at the concentrations recommended by the manufacturer.

To examine the potential for differentiation into somatic lineages in directed differentiation assays, we used the StemCell Technologies StemDiff Trilineage Differentiation Kit. WA09 or WA01 cells were separated by flow cytometry and seeded onto Matrigel-coated 8-well chamber slides, then incubated in three differentiation media for 5 days following the manufacturer's instructions. Then the cultures were fixed in 4% paraformaldehyde in PBSA for 15 min, incubated with primary antibodies (rabbit anti-PAX6 polyclonal IgG Biolegend Poly19013; goat anti-T polyclonal IgG R&D AF2085; goat anti-SOX17polyclonal IgG R&D AF1924) for 3 h, then in secondary antibodies (donkey anti-rabbit IgG Alexa Fluor Plus 488 A21206; donkey anti-goat IgG Alexa Fluor Plus 488 A11070, both from Thermofisher) for 1 h followed by counterstaining with DAPI and examination under indirect immunofluorescence. Sufficient cells from each group were counted to attain a 95% confidence interval of <0.05% for the proportion of positive cells. Commercial antibodies were used at the concentrations recommended by the manufacturer.

**Analytical and preparative flow cytometry**. All single-cell analytical sorting was performed on a BD FACSAria III, using the 100 μm nozzle at 20 psi. Single cells were stained and quantified for the following cell surface markers, GCTM-2 and TG30 or GCTM-2, TG30, and EPCAM. Cells were stained in solution using a mixture of GCTM-2 (mouse IgM, neat hybridoma supernatant) and TG30 (anti-CD9, mouse IgG2a, neat hybridoma supernatant) (double stain) and anti-EPCAM-BV421 (BD Cat. #563180) (triple stain). Primary antibodies against GCTM-2 and TG30 were detected using goat anti-mouse IgM-AF647 (A21238) and goat anti-mouse IgG2a-AF488 (A21131), respectively (Life Tech, Carlsbad, CA). Rat anti-mouse IgG2a Secondary Antibody, PE/Cy7 (RMG2a-62, 407107 Biolegend), was used to detect TG30 in experiments that used FUCCI cell lines, mitochondrial membrane potential dyes, or for the Click-iT® EdU Alexa Fluor® 488 Flow Cytometry Assay (Thermo Fisher, Cat. #10425). Commercial antibodies were used at the concentrations recommended by the manufacturer.

Control samples included unlabeled cells, cells labeled with secondary antibody only and single fluorochrome labeled cells. Cells were sorted using a FACSAria (BD Biosciences) with a 100 μM nozzle and low-pressure conditions. Cells were first gated based on forward and side scatter properties then were analyzed for levels of GCTM-2, TG30, and EPCAM labeling. Double stained cells (GCTM-2 and TG30) were sorted into several populations: GCTM-2$^{low}$CD9$^{low}$, GCTM-2$^{high}$CD9$^{high}$, or GCTM-2$^{high}$CD9$^{high}$EPCAM$^{high}$. The low population consists of cells with low (bottom 25%) expression of GCTM-2 and TG30, the GCTM-2$^{high}$CD9$^{high}$ subset consists of the top 25% of cells expressing GCTM-2 and TG30, whereas the GCTM-2$^{high}$CD9$^{high}$EPCAM$^{high}$ subset is a fraction of the GCTM-2$^{high}$CD9$^{high}$

population with the highest expression of EPCAM, representing ~10% of GCTM-2$^{high}$CD9$^{high}$ fraction. Sorted single cells were processed for gene expression analysis as described below. In some experiments GCTM-2$^{high}$CD9$^{high}$EPCAM$^{high}$ cells were compared with an unsorted (general) population.

For TMRM or JC-1 analysis, rat anti-mouse IgG2a Secondary Antibody, PE/Cy7 (Biolegend 407107, RMG2a-62) was used to detect TG30. $1 \times 10^6$ single cells that were stained with antibodies were resuspended in 1 mL of mTeSR1 supplemented with 250 nM TMRM (Thermo Fisher, Cat. #T668) or 0.3 μg/mL JC-1 (Thermo Fisher, Cat. #T3168) and incubated in the incubator for 30 min. The media was subsequently removed and the cells were washed three times with mTeSR1 before analysis on the flow cytometer.

For Edu incorporation, rat anti-mouse IgG2a Secondary Antibody, PE/Cy7 (Biolegend, 407107, RMG2a-62) was used to detect TG30. Stained single cells were subsequently labeled with Click-iT® EdU Alexa Fluor® 488 Flow Cytometry Assay kit (Thermo Fisher, Cat. #10425) according to manufacturer's method.

All FACs plots were created using FCS express (De Novo Software) and the coefficient of variation was also calculated using FCS express or using the FloJo software package.

**Immunofluorescence microscopy**. For staining of cells in colony formation assays, H9 cells that were cultured in 96-well matrigel-coated plates were washed twice with PBS prior to fixation with 2% paraformaldehyde (PFA) for 30 min at room temperature. Cells were permeabilized with 0.3% Triton X-100 in PBS and blocked with 1% IgG-free BSA, incubated in antibody GCTM-2 overnight at 4 °C, followed by goat anti-mouse IgM Alexa Fluor 488 for 30 min. Samples were then washed with PBS and counterstained with DAPI (Life Tech, Cat. #D1306).

For co-staining of live cells, anti-CD9 primary antibody was preincubated with with goat anti-mouse IgG2a-AF488 (A21131). WA09 hPSC were incubated with mTeSR1 containing 250 nM TMRM (Thermo Fisher, Cat. #T668) and AF488 bound TG30 for 30 min at 37 °C and the cells were then washed three times with warmed mTeSR1 before they were visualized under indirect fluorescence microscopy.

**Metabolomics studies**. The GCTM-2$^{high}$CD9$^{high}$EPCAM$^{high}$ subpopulation or unsorted cells were plated in 6-well format and allowed to expand for 24 h. Spent media was aspirated, and cell cultures were washed once in Milli-Q water to remove extraneous media, then sufficient liquid nitrogen was added to cover the base of the culture surface and enable metabolic arrest. Next, cells were incubated with 600 μl (per 10 cm$^2$ surface area) of ice cold 9:1 MeOH:CHCl$_3$ containing 0.83 μM $^{13}C_6$-sorbitol and 8.3 μM $^{13}C_5$,$^{15}$N-valine for 10 min on ice. The cell lysate was scraped and then transferred to a clean tube, incubated on ice for another 5 min, and centrifuged at $16,100 \times g$ for 5 min at 4 °C following which the supernatant was collected for mass spectrometric analysis.

For stable isotope labeling studies, cells were prepared as described above, but the culture media was supplemented with U-$^{13}C_6$-glucose for 24 h prior to cell harvest.

**LC–MS intracellular metabolite profiling analysis**. Metabolite analysis was performed by LC–MS, using hydrophilic interaction (HILIC) LC and high resolution QTOF mass spectrometry. Sample extracts (10 μl) were injected onto an Agilent 1290 LC fitted with a ZIC-pHILIC column (5 μm, 2.1 × 150 mm; Merck), and 20 mM ammonium carbonate (A) and acetonitrile (B) as the mobile phases. A 14 min gradient starting from 90% B to 40% B over 12 min, held for 2 min followed by washing at 5% B for 3 min and re-equilibration at 90% B, was used. Mass spectrometry utilized an Agilent 6545 QTOF with heated electrospray source operating in negative ionization mode and scan range $m/z$ 50–1700. Conditioning was performed before each batch using 2–3 blanks and 5 mixtures of authentic standards (234 metabolites), which were analyzed in data-dependent MS/MS mode to facilitate downstream metabolite identification where necessary. PBQC samples were analyzed periodically throughout the analysis.

**GC–MS intracellular metabolite profiling analysis**. Five hundred microliters of the cell extract was evaporated to complete dryness under vacuum (Christ RVC 2-33). Polar metabolites were derivatised online using a Gerstel MPS2 XL auto-sampler robot (Gerstel, Germany). Samples were first methoxyaminated by the addition of 20 μL methoxyamine (30 mg/mL in pyridine, 2 h, 37 °C, 750 rpm), followed by trimethylsilylation with 20 μL BSTFA + 1% TMCS (1 h, 37 °C, 750 rpm). Metabolite profiles were acquired on an Agilent 7890A Gas Chromatograph coupled to a 5975C Mass Selective Detector, where 1 μL of derivatised sample was injected into a split/splitless inlet set at 250 °C. Chromatographic separation was achieved using an Agilent VF-5 ms capillary column (30 m × 0.25 mm × 0.25 μm + 10 m duraguard). Oven conditions were set at 35 °C starting temperature, held for 2 min, then ramped at 25 °C/min to 325 °C and held for 5 min. Helium was used as the carrier gas at a flow rate of 1 mL/min. Compounds were fragmented by electron impact (EI) ionization and detected across a $m/z$ range of 50–600 amu, with a scan speed of 9.2 scans/s. Chromatograms were processed using PyMS[58] to align metabolites and quantify a representative target ion, and subsequently generate a data matrix. Metabolites were annotated using the in-house Metabolomics Australia (MA_25C) metabolite library and NIST11 database.

**GC–MS intracellular stable isotope incorporation analysis**. Stable isotope labeled samples were prepared for GC–MS and analyzed as described above. [13]C-targeted metabolomics was performed as previously described[59]. GC–MS was carried out using an Agilent 7890 GC system, VF-5 capillary column with 10 m inert ezi-guard (J&W Scientific, 30 m, 250 µm inner diameter, 0.25 µm film thickness) and an Agilent 5975 MSD (Agilent Technologies, Santa Clara, USA) in electron ionization (EI) mode. We used mass isotopomer peak shift analysis to measure [13]C-glucose derived carbon labeling in key metabolites of the glycolytic and TCA cycle. Elution times and the fragmentation can be found on the NIST database.

**Seahorse flux analysis**. The oxygen consumption rate and extracellular acid-ification rate were determined using an extracellular flux analyzer (Seahorse XFe96 Analyzer, Agilent). GCTM-2[high]CD9[high]EPCAM[high] cells and the unsorted population were seeded into CellTak Cell and Tissue Adhesive (Corning)-coated wells of a Seahorse XF96 Cell Culture Microplate ($n > 6$) at $10^5$ cells per well in 180 µl Seahorse XF Assay Medium (supplemented with 10 mM glucose, 2 mM gluta-mine and 1 mM sodium pyruvate). The mitochondrial electron transport chain was challenged using the Seahorse XF Mito Stress Test Kit (Agilent). A total of 12 measurement cycles were carried out, each cycle consisting of 3 min mixing and 3 min measurement of the oxygen consumption rate and extracellular acidification rate. The first three cycles determined the basal rate and $3 \times 3$ additional cycles were measured after addition of oligomycin (2 µM final), carbonyl cyanide-4-(trifluoromethoxy)phenylhydrazone (FCCP, 2 µM final) and rotenone/antimycin A (0.5 µM final each). The data were exported and analyzed using Excel (Microsoft) and Wave Software (Agilent).

**Reduced representational bisulfite sequencing**. FACS sorted cells were snap frozen in buffer RLT plus (Qiagen) before DNA extraction using the AllPrep DNA/RNA Mini Kit (Qiagen) as per the manufacturer's instructions. DNA was treated with RNase A then purified through the DNA Clean and Concentrator column (Zymo). RRBS libraries were made from 100 ng of purified DNA using the Ovation RRBS Methyl-Seq System (NuGEN), according to the manufacturers recommen-dations, which includes the Qiagen Epitect kit for bisulfite conversion. Sequencing was performed on a NextSeq 500 with a 75 bp single-end sequencing protocol[60]. Sequence quality control was performed using FastQC[61]. Trimming of adapters and low-quality base calls was performed with trim_galore[62]. Trimmed reads were filtered for true RRBS reads (which contain an *Msp*I cut site at the 5′ using trimRRBSdiversityAdaptCustomers.py (NuGEN). Reads were aligned to a bisulfite converted human genome (hg38), using Bismark[63] Methylation calls were made with bismark_methylation_extractor[63]. Analysis of methylation over CpG islands was performed using Seqmonk[64] where only CGIs with 10 or more informative CpG sites were considered. FastQC, trim_galore, and Seqmonk are all available from www.bioinformatics.babraham.ac.uk.

**Chromatin accessibility**. DNA accessibility was measured using the FAST-ATAC protocol[65] with 50,000 sorted cells for each of three biological replicates of both GCTM-2[high]CD9[high]EPCAM[high] and GCTM-2[mid]CD9[mid] populations. Sorted cells were pelleted by centrifugation for 5 min at 4 °C, supernatant was removed, and the cell pellet was washed one time with PBS pH 7.2. Cells were resuspended in 50 µl of transposase mixture consisting of 25 µl of 2x TD buffer (Illumina), 2.5 µl of TDE1 enzyme (Illumina), 0.5 µl of 1% digitonin (G9441, Promega), and 22 µl of nuclease-free water. Transposase reactions were incubated at 37 °C for 30 min with constant shaking in an Eppendorf ThermoMixer. Following transposition, DNA was pur-ified using the Zymo DNA Clean and Concentrator-5 Kit (#D4014) following manufactures protocol and eluted in 20 µl of 10 mM Tris-HCL, pH 8. Transposase samples were barcoded using duel indexes during library amplification using NEBNext Ultra II Q5 Master Mix (#M0544L), 25 µM of each primer, and 20 µl of transposed DNA. Libraries were amplified for a total of nine cycles (98 °C for 10 s, 63 °C or 30 s, 72 °C for 1 min), followed by purification using AMPure beads (Beckman Coulter #A63881). ATAC-seq libraries were visualized on Bioanalyzer for typical nucleosome banding followed by sequencing on the Illumina Next-Seq platform. Libraries were trimmed using trimmomatic[66] and aligned to Ensemble (build hg19) using bwa[67] with default settings. Duplicates reads were removed using Picard tools MarkDuplicates, and each aligned read was shifted toward the Tn5 cut site as described[37]. As reported, we found that FAST-ATAC resulted in low percentages of reads mapping to mitochondria (Supplementary Fig. 4a). Regions of open chromatin were determined by combining alignment files across replicates and using MACS v.1.4.3[68]. Open chromatin regions were merged between both high and low samples to form a universal set of peaks (peakome), and regions from ENCODE blacklist removed. Reads were quantified for each interval in the peakome using bedtools[69] and normalized using TMM method[70]. We found that each sample had a high fraction of reads in the peakome, showed strong enrichment for open chromatin at transcription start sites, and characteristic distribution of fragments showing nucleosome free regions and mono- and di-nucleosome patterns, indicating high-quality ATAC libraries (Supplementary Fig. 4b–d). Data exploration using principle component analysis (PCA) found that the first PCA, explaining 43% of the variance, separated high and low populations, while the second PCA repre-sented potential batch effects (Supplementary Fig. 4e, f). Based on these observations, a general linearized model in edgeR[70] was used to identify differences in DNA

accessibility between populations including batch as a covariate. Peak set annotations and enrichments were calculated using LOLA[39] by comparing peaks more accessible in either high or low populations (FDR < 0.01, Supplementary Table 1) with the universal set of DNAse hypersensitivity sites as the universe against the LOLA core (Supplementary Data 2). Statistical analysis, data exploration, and visualization was performed using R (http://www.R-project.org).

**Single-cell RNA sequencing**. Single cells were flow sorted into a chilled 384-well PCR plate containing 1.2 µl of primer/lysis mix [20 nM indexed polydT primer, 1:6,000,000 dilution of ERCC RNA spike-in mix (Ambion, 4456740), 1 mM dNTPs, 1.2 units SUPERaseIN Rnase Inhibitor (Thermo Fisher, AM2696), DEPC water (Thermo Fisher, AM9920)] using a BD FACSAria III flow cytometer (BD Biosciences, San Jose, CA, USA) and the protocol described above. Sorted plates were sealed, centrifuged for 1 min at 3000 rpm and immediately frozen upside down at −80 °C until further processing using an adapted CelSeq2 protocol[71].

Sorted plates were thawed on ice and briefly centrifuged. To lyse the cells and anneal the mRNA capture primer the plate was incubated at 65 °C for 5 min and immediately chilled on ice for at least 2 min before adding 0.8 µl reverse transcription reaction mix [in 2 µl RT reaction: 1x Fist Strand buffer (Invitrogen, 18064-014), 20 mM DTT (Invitrogen, 18064-014), 4 units RNaseOUT (Invitrogen, 10777-019), 10 units SuperScript II (Invitrogen, 18064-014)]. The plate was incubated at 42 °C for 1 h, 70 °C for 10 min and chilled to 4 °C to generate first strand cDNA. For second strand cDNA synthesis 6 µl of second strand reaction mix were added [1x NEBNext Second Strand Synthesis buffer (NEB #E6111S), NEBNext Second Strand Synthesis Enzyme Mix: 2.4 units DNA Polymerase I (*E. coli*), 2 units RNase H, 10 units *E. coli* DNA Ligase (NEB #E6111S), DEPC water (Thermo Fisher, AM9920)]. The plate was incubated at 16 °C for 2 h to generate double stranded cDNA.

All samples were pooled and cleaned using a 1.2X NucleoMag NGS Clean-up and Size select magnetic beads (Macherey-Nagel, 7449970.5) according to manufactures instruction. To reduce the amount of beads for each 100 µl pooled sample, 20 µl beads and 100 µl bead binding buffer (20% PEG8000, 2.5 M NaCl, pH 5.5) was added. The cDNA was eluted in 6.4 µl DEPC water and kept with beads for the following IVT reaction were 9.6 µl of IVT reaction mix [1.6 µl of each of the following: A, G, C, U, 10X T7 buffer, T7 enzyme (MEGAscript T7 transcription kit (Ambion, AM1334))] was added and incubated at 37 °C for 13 h and then chilled and kept at 4 °C. To remove the leftover primers, 6 µl ExoSAP-IT For PCR Product Clean-Up (Affymetrix, 78200) was added and the sample was incubated at 37 °C for 15 min and then chilled and kept at 4 °C.

Chemical heat fragmentation was performed by adding 5.5 µl of 10X Fragmentation buffer (RNA fragmentation reagents, AM8740) to the sample and incubation in pre-heated thermal cycler at 94 °C for 2.5 min followed by immediately chill on ice and addition of 2.75 µl of Fragmentation Stop buffer (RNA fragmentation reagents, AM8740). The fragmented amplified RNA was purified using 1.8X RNAClean XP beads (Beckman Coulter, A63987) according to manufactures instruction and eluted in 6 µl DEPC water of which 5 µl (no beads) were transferred to a fresh tube for library preparation.

The fragmented RNA was transcribed into cDNA using 5′-tagged random hexamer primers (GCCTTGGCACCCGAGAATTCCANNNNNN) introducing a partial Illumina adapter as also described in ref. [71]. To remove RNA secondary structure and anneal the mRNA capture primer 1 µl of tagged random hexamer (100 µM) and 0.5 µl of 10 mM dNTPs (dNTP solution set NEB, N0446S) were added to the sample and incubated at 65 °C for 5 min and immediately chilled on ice for at least 2 min before adding 4 µl reverse transcription reaction mix [in 10 µl RT reaction: 1x First Strand buffer (Invitrogen, 18064-014), 20 mM DTT (Invitrogen, 18064-014), 4 units RNaseOUT (Invitrogen, 10777-019), 10 units SuperScript II (Invitrogen, 18064-014)].

The PCR primers introduce the full-length adaptor sequence required for Illumina sequencing (for details see Illumina small RNA PCR primers). PCR was performed in 12.5 µl using half of the ranhexRT sample as a template [1X KAPA HiFi HotStart ReadyMix (KapaBiosystems KK2602), 400 nM each primer].

The final PCR amplified Library was submitted to two consecutive 1x NucleoMag NGS Clean-up and Size select magnetic beads (Macherey-Nagel, 7449970.5) according to manufactures instruction. The final library was eluted in 20 µl of 10 mM Tris-HCl solution (Sigma-Aldrich, T2319-1L).

**RNA-seq data analysis**. CelSeq2 scRNA-sequencing reads were mapped to the GRCh38 human genome using the Subread aligner[72] and assigned to genes using scPipe[73] with ENSEMBL v86 annotation. Gene counts were generated as a matrix by scPipe with UMI-aware counting and imported into R. Cells were removed from further analysis if they fewer than 12,500 total counts, or >60,000 total counts, or 4500 total genes detected. Genes were filtered out if they failed to achieve 1 count in at least 10% of a particular cell condition group. Heatmaps were generated on normalized expression values using heatmap2 from the gplots package with row normalization. Dimensionality reduction was performed on normalized log2-cpm expression values with size factors from computeSumFactors in scran[74]. Single-cell RNA-seq data are available through the Gene Expression Omnibus under accession number #:GSE119323.

The non-human primate data were generated on the SC3-seq platform[75] and are publicly available under GEO accession number GSE74767. In order to

compare between human and macaque expression levels, we defined a set of high confidence orthologous metaexons between the two species as in ref. [76]. Briefly, we used BLAT[77] to compare every annotated human exon in ENSEMBL release 86 (737,982 unique exons across 63,305 genes) against the human (hg38) and Macaca fascicularis (macFas5) genomes, retaining those that matched the macaque genome with at least 92% sequence identity and mapped back to their annotated location in humans. We then excluded all exons that had a second match with >90% sequence similarity in either genome, to control for interspecies differences in mappability. Overlapping exons from the same gene (associated with different isoforms) were collapsed into a single "metaexon".

We discarded overlapping exons associated with more than one ENSEMBL gene ID, exons associated with any gene annotated to two or more chromosomes in either species, and exons where the difference in intron size between the two species is ≥10,000 bp, suggestive of poor genome assembly or annotation. After applying these quality control criteria, we ultimately retained 198,172 unique metaexons in humans and macaques across 34,142 annotated ENSEMBL human genes. The final table, as well as code and additional documentation for metaexon identification is available at http://www.bitbucket.org/ee_reh_neh/orthoexon

We processed all 2526 files from ref. [44] (from 390 cells) with sickle [https://github.com/najoshi/sickle] to remove bad reads and trim low-quality bases from the 3′ end. We then mapped all reads to macFas5 using Rsubread 1.20.6 and R 3.2.2, allowing up to 2 mismatches and 2 indels per 50 bp read, which is proportional to our setting of 5 mismatches or indels for a 100 bp read. Mapped reads were assigned to the orthologous metaexon list using featureCounts at both the single metaexon and whole-gene level, and summed within individuals in R 3.2.2.

**Human scRNA-seq data processing**. Sequence analysis was performed on the Illumina Next-Seq 500 platform.

Quality control of raw data was assessed using FASTQC and visualized using MultiQC. The scPipe package v1.0 for R was used to count genes based on UMI profile. Gene expression was normalized using scater v1.6.1 and scran v1.6.6 packages for R. FASTQ files were aligned to hg38 using the Subread package v1.26.1 for R statistical software, aligned reads were re-annotated to exons using ENSEMBL v86 transcriptome to define the exon/intron mapping rate. Cells with <4500 expressed genes or more than 60,000 counts were discarded, resulting in 300 of 370 non-control cells retained for downstream analysis.

Data from both species reflected Log2(CPM + 1) aligned to HG38v86. Data from both species were mapped to a set of highly orthologous metaexons. Supplementary Fig. 8 illustrates the distribution of gene expression in each dataset before and after gene-based filtering. A gene was retained in a given dataset based on expression above a cut off of 1 (non-human primate) or 5 (human) Log2(CPM + 1) in at least 10% of cells assigned to a given phenotype. These cutoffs were selected based on the distribution of gene expression in each dataset. Downstream analysis was limited to the 7308 genes commonly expressed in each dataset.

To rescale and combine the datasets, each gene was assigned a rank in a given cell based on its abundance. Ties in the data were assigned the same rank and the minimum value was used. Supplementary Fig. 9 illustrates the distributions of $z$-scores of ranked gene expression for every cell in each dataset. PCA was performed on the merged data using prcomp function in the stats base package for R statistical software version 3.3.2. Downstream analysis of the principal components was performed using the mixOmics package version 6.3.1 for R version 3.3.2.Panther.db was used to perform Fischer's exact test for over-representation of ontological terms in gene sets of interest.

**Reporting summary**. Further information on research design is available in the Nature Research Reporting Summary linked to this article.

## Data availability
RNA-seq, scRNA-seq, and ATAC-seq data are available at Gene Expression Omnibus under accession Superseries GSE119326 (RNA-seq, GSE119324, scRNA-seq GSE119323, ATAC-seq GSE147338).

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

## Acknowledgements

This study was supported by the Australian Research Council Special Research Initiative in StemCell Sciences (SR1101002), and Bioplatforms Australia, through funding from the Australian Government National Collaborative Research Infrastructure Strategy. M.J.M. is a Principal Research Fellow of the National Health and Medical Research Council. M.E.B. was supported by a Bellberry-Viertel Senior Medical Research Fellowship.

## Author contributions

Contributions of authors to drafting of specific sections of the paper are identified by name of section titles. K.X.L.: design of experiments, collection of data, and writing of the paper (Methods); E.A.M.: analysis of data and writing of the paper (Methods); J.K.: design of experiments, collection of data, and writing of the paper (Methods); D.P.D. and M.M.: design of experiments, analysis of data, and writing of the paper (Methods, Results); J.K.: collection and analysis of data; D.T., T.B., C.S., and T.C.M.: collection of data; A.K.: design of experiments, analysis and collection of data, and writing of the paper (Methods); M.E.B.: design of experiments, analysis of data; M.E.R.: analysis of data; S.H.N.: design of experiments, analysis of results, writing of the paper (Methods); D.Z.: design of experiments and collection of data; O.K. and S.S.: analysis of results; I.G.R., design of experiments, analysis of results, writing of manuscript (Methods); C.L.B.: design of experiments, analysis of results, writing of the paper; C.A.W.: design of experiments, analysis of results, and writing of the paper (Methods, Results); M.F.P.: conception of study, design of experiments, analysis of results, and writing of the paper.

## Competing interests

The authors declare no competing interests.
