## [Peer Review File · Nature Communications]

Editorial Note: This manuscript has been previously reviewed at another journal that is not operating a transparent peer review scheme. This document only contains reviewer comments and rebuttal letters for versions considered at Nature Communications .

Reviewers' comments:

Reviewer #1 (Remarks to the Author):

The authors have now addressed most previous criticism. Fig.2 however needs further work. The Germline differentiation potential data of hPSC are described in line 168-191 and show increased PGC-like cell differentiation capacity in the HHH group. However, these data are very hard to extract from the Fig.2. Please add the data described in the text to Fig.2 and/or clarify Fig.2. The authors describe high degree of inter-assay variability. How much do the repeats in the same assay vary (intra-assay variability)? Furthermore, does HHH population have normal Karyotype?

Reviewer #2 (Remarks to the Author):

The authors have done a good job of addressing comments made from the previous review. This is potentially important work but I would like to see some further characterization of differentiation potential for 'archetypical' cells to show that they are really pluripotent. This is critical.

Specific comments

1. Can 'archetypical' cells be converted to naive/EPC/primed cells? This would help define where these cells are with respect to other PSCs.

2. Can 'archetypical' cells be differentiated to the three germ layers? Are they pluripotent and how does their potential compare to primed cells?

Is it possible that 'archetypical' cells have a reduced capacity for germ layer formation while having increased germ cell capacity?

3. The authors probably feel boxed in by existing nomenclature (ground-state, naive etc) but perhaps this would be a good opportunity to suggesting a different nomenclature. I don't really like 'archetypical' as it isn't informative and is an inappropriate use of the word- maybe "founder PSCs"; "fPSCs" would be better terminology?

Reviewer #3 (Remarks to the Author):

The revised manuscript addresses in part issues raised for the previous submission. The authors have further characterised the sub-population of cells within human pluripotent stem cell cultures that have higher self-renewal potential. Although several questions remain open, the ATACseq analyses are a useful addition that substantiates the conclusion that this sub-population differs in molecular properties from the bulk population and may represent an earlier developmental stage. On balance, the observations in this study are appropriate for publication due to the fundamental interest and importance in describing accurately the status and identity of human pluripotent stem cell cultures. The topicality is illustrated by two recent papers that are published in high profile journals (Nakanishi et al., Cell; Cornacchia et al., Cell Stem Cell) but are far from definitive and will be well complemented by the Lau et al study. However, some results (PGC differentiation, methylome analysis, cell cycle analysis) are inconclusive and could be relegated to supplemental with reduced main text description. Indeed the manuscript is in general overlong, which rather detracts from the clarity of the authors' main message.

Suggestions to improve the manuscript:

Please be explicit at the beginning of the results about the culture medium. I surmise this is mTeSR but it is not stated. The methods mention E8 but do they find the same population heterogeneity in E8. This issue must be clearly addressed because E8 has become widely adopted and is purported to maintain more homogeneous cultures.

Following on from the above, the authors should compare their findings with the recent paper by Cornacchia et al. reporting that hPSCs in E8 have distinct metabolic features and represent an intermediate stage of pluripotency. This is particularly pertinent to the metabolomic analyses. Please clarify in the text what is meant by GEN. Is this the bulk population or the population after sorting ESR cells?

The PGC cell surface markers used are also expressed by hPSC. The claim of PGC-like identity requires demonstration of PGC specific markers such as NANOS3. In any case I do not see what this experiment adds to the paper since it is well established that "archetypal" hPSC can be manipulated into PGCLC differentiation and the significance of small quantitative differences is unclear when there is such inherent variability.

The methylome analyses are rather superficial and add nothing beyond the statement that there are no major differences between ESR and the rest of the population. The claim that there are low levels of methylation is not well-supported and the anecdotal comparison with mouse EpiSC is not justified by any analysis.

The cell cycle data presented are potentially misleading because the authors have compared HH cells with LL cells, but not with MM cells as in the self-renewal assays. The authors should either repeat the experiment with the proper comparison or remove these data. Furthermore, although argued by some, it is absolutely not clear that lineage commitment decisions occur in G1.

The statement that ESR cells "show a pattern of gene expression that is strongly similar to peri-implantation epiblast stages in the primate embryo, but clearly distinguished from inner cell mass or gastrulation stages" should be toned down and phrased more accurately to reflect the post-implantation (not peri-implantation) stages of primate epiblast examined.

Please stated clearly in the text the proportion of HH cells in a typical culture and how variable this is. The authors must clarify whether LL cells are primed, as the data suggest but they argue against in the response letter.

We thank the Reviewers for their favorable comments on our revision and for highlighting a few additional issues that need attention. We have addressed the remaining points through additional experimentation and through revision of the text, and by providing more explanation for the referees where indicated. Our detailed replies to the Reviewers are below. Comments to the Reviewers are in blue; notations of alterations to the manuscript and figures are in Red. In the Revised manuscript, changes to the text in the previous revision are in red, and changes in the current revision are in purple.

Reviewer #1 (Remarks to the Author):

The authors have now addressed most previous criticism. Fig.2 however needs further work. The Germline differentiation potential data of hPSC are described in line 168-191 and show increased PGC-like cell differentiation capacity in the HHH group. However, these data are very hard to extract from the Fig.2. Please add the data described in the text to Fig.2 and/or clarify Fig.2. The authors describe high degree of inter-assay variability. How much do the repeats in the same assay vary (intra-assay variability)? Furthermore, does HHH population have normal Karyotype?

We have reformatted Figure 2 to clarify it. We include the data described in the text in a table form (within an assay results were quite consistent) and we also show staining with PRDM1 and NANOS3 to confirm the identity of the differentiated cells (see query from Reviewer 3). We also show data on embryoid body differentiation to confirm our previous work showing that the HHH population is pluripotent (see query from Reviewer 2). As we noted in the previous version, the cell lines used show normal karyotypes (Online Methods, l 647-648). Revised Figure 2; text Results l 184-186 and 188-192, Online Methods l 699-716, Figure Legends l 1304-1305, 1310-1318.

Reviewer #2 (Remarks to the Author):

The authors have done a good job of addressing comments made from the previous review. This is potentially important work but I would like to see some further characterization of differentiation potential for 'archetypical' cells to show that they are really pluripotent. This is critical.

Specific comments

1. Can 'archetypical' cells be converted to naive/EPC/primed cells? This would help define where these cells are with respect to other PSCs.

The ESR subpopulation can reconstitute the entire population including primed cells, as shown by (Polanco et al., 2013) and in Figure 1c. We have not investigated whether these cells can be converted to the naive state.

2. Can 'archetypical' cells be differentiated to the three germ layers? Are they pluripotent and how does their potential compare to primed cells?

Is it possible that 'archetypical' cells have a reduced capacity for germ layer formation while having increased germ cell capacity?

We have shown previously that the ESR subpopulation is pluripotent (Figures 4 and 5 (Hough et al., 2014)). We include new data from embryoid body assays to confirm this point. Revised Figure 2D; text, Results 188-192, Online Methods 1 705-716, Figure Legends 1 1316-1318.

3. The authors probably feel boxed in by existing nomenclature (ground-state, naive etc) but perhaps this would be a good opportunity to suggesting a different nomenclature. I don't really like 'archetypical' as it isn't informative and is an inappropriate use of the word- maybe "founder PSCs"; "fPSCs" would be better terminology? We appreciate the difficulty. We use archetypal in the sense of its dictionary definition (archetype meaning prototype), since this hPSC phenotype represents a stable attractor state that is displayed by cell lines made from germ cell tumors, embryos, or through reprogramming, under a variety of cell culture conditions.

Reviewer #3 (Remarks to the Author):

The revised manuscript addresses in part issues raised for the previous submission. The authors have further characterised the sub-population of cells within human pluripotent stem cell cultures that have higher self-renewal potential. Although several questions remain open, the ATACseq analyses are a useful addition that substantiates the conclusion that this sub-population differs in molecular properties from the bulk population and may represent an earlier developmental stage. On balance, the observations in this study are appropriate for publication due to the fundamental interest and importance in describing accurately the status and identity of human pluripotent stem cell cultures. The topicality is illustrated by two recent papers that are published in high profile journals (Nakanishi et al., Cell; Cornacchia et al., Cell Stem Cell) but are far from definitive and will be well complemented by the Lau et al study. However, some results (PGC differentiation, methylome analysis, cell cycle analysis) are inconclusive and could be relegated to supplemental with reduced main text description. Indeed the manuscript is in general overlong, which rather detracts from the clarity of the authors' main message. Please see the responses below to questions/suggestions regarding specific sections of the results.

Suggestions to improve the manuscript:

Please be explicit at the beginning of the results about the culture medium. I surmise this is mTeSR but it is not stated. The methods mention E8 but do they find the same population heterogeneity in E8. This issue must be clearly addressed because E8 has become widely adopted and is purported to maintain more homogeneous cultures. Experiments were run in mTeSR or E8 (Online Methods 1 645-648). These media are essentially equivalent in terms of their capacity to support the ESR; both maintain a population of cells that has very few GCTM-2/CD9 double negative cells, and the proportion of HHH cells is similar in both.

Following on from the above, the authors should compare their findings with the recent paper by Cornacchia et al. reporting that hPSCs in E8 have distinct metabolic features and represent an intermediate stage of pluripotency. This is particularly pertinent to the metabolomic analyses. Thanks for drawing our attention to this recent paper from Lorenz Studer's group. The study of Cornacchia et al. (Cornacchia et al., 2019) is entirely in line with what we and others have observed here and elsewhere (Hough et al., 2014; Kolle et al., 2009), namely that culture in E8 (or mTeSR) relative to serum or serum replacer shifts the population towards the HHH fraction but does not entirely eliminate heterogeneity. We cite this study of Cornacchia et al. and note this interpretation in the revised Discussion. **Revised Discussion I 595-604, References I 1226-1228**

Please clarify in the text what is meant by GEN. Is this the bulk population or the population after sorting ESR cells? **This point was clarified in the previous revision; it represents the remaining cells not in the ESR fraction.**

The PGC cell surface markers used are also expressed by hPSC. The claim of PGC-like identity requires demonstration of PGC specific markers such as NANOS3. In any case I do not see what this experiment adds to the paper since it is well established that “archetypal” hPSC can be manipulated into PGCLC differentiation and the significance of small quantitative differences is unclear when there is such inherent variability. **Indeed these markers are expressed by hPSC, but note that in this protocol EPCAM⁺ cells disappear after the mesoderm stage and only reappear in cultures that have received cytokines that induce PGC-like cells. We show staining with PRDM1 and NANOS3 in Revised Figure 2 to confirm PGC-like identity. We carried out this experiment to confirm that the ESR studied here was in fact capable of germ cell differentiation, in contrast to mouse primed epiblast stem cells. In previous studies of heterogeneous cultures, the precise population of cells giving rise to PGC was not clear. These results and the confirmatory embryoid body assay (see response to Reviewer 2) show that indeed ESR have the developmental potential of early post-implantation epiblast. Revised Figure 2; text Results I 184-186 and 188-192, Online Methods I 699-716, Figure Legends I 1304-1305, 1310-1318.**

The methylome analyses are rather superficial and add nothing beyond the statement that there are no major differences between ESR and the rest of the population. The claim that there are low levels of methylation is not well-supported and the anecdotal comparison with mouse EpiSC is not justified by any analysis. **We hope that we have not overinterpreted these data; we do not claim overall low levels of DNA methylation but only that a significant number of loci in both HHH and the remaining cells are hypomethylated, in contrast with mouse EpiSC. The data do support this point.**

The cell cycle data presented are potentially misleading because the authors have compared HH cells with LL cells, but not with MM cells as in the self-renewal assays. The authors should either repeat the experiment with the proper comparison or remove these data. Furthermore, although argued by some, it is absolutely not clear that lineage commitment decisions occur in G1. **The point of this experiment was to show that HHH cells in defined conditions have a very low G0 or G1 fraction, relative to the GEN population, and thus differ from primed cells. For cell cycle data on MM versus HH cells see our previous work (Filipczyk et al., 2007). We acknowledge that there is some difference of opinion regarding the G1 lineage specification decision point but certainly evidence to**

support it; resolution of this issue is beyond the scope of this manuscript.

The statement that ESR cells “show a pattern of gene expression that is strongly similar to peri-implantation epiblast stages in the primate embryo, but clearly distinguished from inner cell mass or gastrulation stages” should be toned down and phrased more accurately to reflect the post-implantation (not peri-implantation) stages of primate epiblast examined. There are some features of gene expression in the HHH subpopulation that are similar to pre-implantation primate epiblast; this is reflected in Figures 8 and 9. However, we do not wish to claim that this subpopulation represents a pre-implantation epiblast or naive state. So we have used the term early post-implantation throughout this revision, though the transition between pre and post implantation epiblast may be less abrupt than in the mouse. Results | 357, 469 Discussion | 595.

Please stated clearly in the text the proportion of HH cells in a typical culture and how variable this is. Please see Methods | 755-763. For a given set of culture conditions the proportions are very reproducible though as noted, defined conditions enhance the HH and HHH fractions.

The authors must clarify whether LL cells are primed, as the data suggest but they argue against in the response letter. We are not sure what the Reviewer is referring to here, cells outside of the ESR show neural lineage priming as discussed.

Cornacchia, D., Zhang, C., Zimmer, B., Chung, S.Y., Fan, Y., Soliman, M.A., Tchieu, J., Chambers, S.M., Shah, H., Paull, D., *et al.* (2019). Lipid Deprivation Induces a Stable, Naive-to-Primed Intermediate State of Pluripotency in Human PSCs. *Cell Stem Cell* 25, 120-136 e110.

Filipczyk, A.A., Laslett, A.L., Mummery, C., and Pera, M.F. (2007). Differentiation is coupled to changes in the cell cycle regulatory apparatus of human embryonic stem cells. *Stem Cell Res* 1, 45-60.

Hough, S.R., Thornton, M., Mason, E., Mar, J.C., Wells, C.A., and Pera, M.F. (2014). Single-cell gene expression profiles define self-renewing, pluripotent, and lineage primed States of human pluripotent stem cells. *Stem cell reports* 2, 881-895.

Kolle, G., Ho, M., Zhou, Q., Chy, H.S., Krishnan, K., Cloonan, N., Bertinello, I., Laslett, A.L., and Grimmond, S.M. (2009). Identification of human embryonic stem cell surface markers by combined membrane-polysome translation state array analysis and immunotranscriptional profiling. *Stem Cells* 27, 2446-2456.

Polanco, J.C., Ho, M.S., Wang, B., Zhou, Q., Wolvetang, E., Mason, E., Wells, C.A., Kolle, G., Grimmond, S.M., Bertinello, I., *et al.* (2013). Identification of unsafe human induced pluripotent stem cell lines using a robust surrogate assay for pluripotency. *Stem Cells* 31, 1498-1510.

Reviewers' comments:

Reviewer #1 (Remarks to the Author):

The authors have now adequately addressed the reviewer comments.

Reviewer #2 (Remarks to the Author):

I don't think the characterization of differentiation potential into the three germ layers is sufficient to address the question raised (revised Figure 2d). The data doesn't compare differentiation potential quantitatively and the data presented is of low quality and value. More experimentation is required to address this important point.

The original comment was

"Can 'archetypical' cells be differentiated to the three germ layers? Are they pluripotent and how does their potential compare to primed cells?

Is it possible that 'archetypical' cells have a reduced capacity for germ layer formation while having increased germ cell capacity?"

Reviewer #3 (Remarks to the Author):

No further comments.

We thank the referees for their positive assessments of our study. We understand the remaining concerns of Reviewer 2:

I don't think the characterization of differentiation potential into the three germ layers is sufficient to address the question raised (revised Figure 2d). The data doesn't compare differentiation potential quantitatively and the data presented is of low quality and value. More experimentation is required to address this important point.

The original comment was

"Can 'archetypal' cells be differentiated to the three germ layers? Are they pluripotent and how does their potential compare to primed cells?

Is it possible that 'archetypal' cells have a reduced capacity for germ layer formation while having increased germ cell capacity?"

As we pointed out in our previous response to the reviewer, we have shown elsewhere, using both indirect immunofluorescence of embryoid bodies, and QRT-PCR, that the self-renewing fraction of archetypal hPSC (we use "archetypal" to refer to cultures grown under conventional conditions-primed cells; the reviewer is referring to what we call the self-renewing subpopulation) is capable of differentiation into all three germ layers (Hough et al., 2014). In addition, Andy Laslett's laboratory, using the same FACS methodology as ours, demonstrated pluripotency of this subpopulation through the teratoma assay (Polanco et al., 2013). Moreover, as indicated in Figure 1 a and c of this study (and noted in a previous rebuttal), and shown more extensively by Laslett's laboratory (Polanco et al., 2013), following isolation and re-plating, the self-renewing fraction *regenerates the broad population of archetypal or primed cells*. Therefore, in accordance with the differentiation data above, it follows that the self-renewing population inherently possesses the same differentiation capacity as the bulk population, since it can regenerate the latter.

Now we appreciate more clearly that the reviewer is requesting a quantitative indication of tri-lineage germ layer differentiation. Though on the face of it this is a valid query, experience teaches that it is difficult to address this request in a definitive manner. Our large scale ISCI comparison of different methods to assess pluripotency, published in your journal (International Stem Cell, 2018), compared EB assays using spontaneous or directed differentiation, or teratoma assays, in their ability to assess pluripotency. This multi-institutional study showed quite clearly that while all three assays could provide *qualitative* information on the ability of a given cell to differentiation along the three germ layer lineages, the *quantitative outcome* of trilineage differentiation is assay dependent (see figure reproduced from this study below showing scorecard gene expression outcomes in spontaneous versus directed differentiation EB assays). In other words, it is not possible to reach generalizable quantitative conclusions about propensity for germ layer lineage differentiation from a particular assay. Quantitative outcomes are assay context-dependent, and there is no one definitive assay for this purpose.

Nevertheless, to respond to this last outstanding issue, we have now carried out studies of directed differentiation in adherent culture to compare the developmental potential of the ESR fraction with the remaining population in two cell lines, with enumeration of cells positive for markers of all three germ layer lineages. The results (revised Figure 2d) show clearly that both the ESR and the remaining population are pluripotent, in accordance with the previous studies above and our embryoid body differentiation study in the last revision.